# PROVABLE FICTITIOUS PLAY FOR GENERAL MEAN-FIELD GAMES

## ABSTRACT

We propose a reinforcement learning algorithm for stationary mean-field games, where the goal is to learn a pair of mean-field state and stationary policy that constitutes the Nash equilibrium. When viewing the mean-field state and the policy as two players, we propose a fictitious play algorithm which alternatively updates the mean-field state and the policy via gradient-descent and proximal policy optimization, respectively. Our algorithm is in stark contrast with previous literature which solves each single-agent reinforcement learning problem induced by the iterates mean-field states to the optimum. Furthermore, we prove that our fictitious play algorithm converges to the Nash equilibrium at a sublinear rate. To the best of our knowledge, this seems the first provably convergent reinforcement learning algorithm for mean-field games based on iterative updates of both mean-field state and policy.

## 1 INTRODUCTION

Multi-agent reinforcement learning (MARL) (Shoham et al., 2007; Busoniu et al., 2008; Hernandez-Leal et al., 2017; Hernandez-Leal et al.; Zhang et al., 2019) aims to tackle sequential decision-making problems in multi-agent systems (Wooldridge, 2009) by integrating the classical reinforcement learning framework (Sutton & Barto, 2018) with game-theoretical thinking (Başar & Olsder, 1998). Powered by deep-learning (Goodfellow et al., 2016), MARL recently has achieved striking empirical successes in games (Silver et al., 2016; 2017; Vinyals et al., 2019; Berner et al., 2019; Schrittwieser et al., 2019), robotics (Yang & Gu, 2004; Busoniu et al., 2006; Leottau et al., 2018), transportation (Kuyer et al., 2008; Mannion et al., 2016), and social science (Leibo et al., 2017; Jaques et al., 2019; Cao et al., 2018; McKee et al., 2020).

Despite the empirical successes, MARL is known to suffer from the scalability issue. Specifically, in a multi-agent system, each agent interacts with the other agents as well as the environment, with the goal of maximizing its own expected total return. As a result, for each agent, the reward function and the transition kernel of its local state also involve the local states and actions of all the other agents. As a result, as the number of agents increases, the capacity of the joint state-action space grows exponentially, which brings tremendous difficulty to reinforcement learning algorithms due to the need to handle high-dimensional input spaces. Such a curse of dimensionality due to having a large number of agents in the system is named as the "curse of many agents" (Sonu et al., 2017).

To circumvent such a notorious curse, a popular approach is through mean-field approximation, which imposes symmetry among the agents and specifies that, for each agent, the joint effect of all the other agents is summarized by a population quantity, which is oftentimes given by the empirical distribution of the local states and actions of all the other agents or a functional of such an empirical distribution. Specifically, to obtain symmetry, the reward and local state transition functions are the same for each agent, which are functions of the local state-action and the population quantity. Thanks to mean-field approximation, such a multi-agent system, known as the mean-field game (MFG) (Huang et al., 2003; Lasry & Lions, 2006a;b; 2007; Huang et al., 2007; Guéant et al., 2011; Carmona & Delarue, 2018), is readily scalable to an arbitrary number of agents.

In this work, we aim to find the Nash equilibrium (Nash, 1950) of MFG with infinite number of agents via reinforcement learning. By mean-field approximation, such a game consists of a population of symmetric agents among which each individual agent has infinitesimal effect over the

whole population. By symmetry, it suffices to find a symmetric Nash equilibrium where each agent adopts the same policy. Under such consideration, we can focus on a single agent, also known as the representative agent, and view MFG as a game between the representative agent's local policy $\pi$ and the mean-field state $\mathcal{L}$ which aggregates the collective effect of the population. Specifically, the representative agent $\pi$ aims to find the optimal policy when the mean-field state is fixed to $\mathcal{L}$, which reduces to solving a Markov decision process (MDP) induced by $\mathcal{L}$. Simultaneously, we aim to let $\mathcal{L}$ be the mean-field state when all the agents adopt policy $\pi$. The Nash equilibrium of such a two-player game, $(\pi^*, \mathcal{L}^*)$, yields a symmetric Nash equilibrium $\pi^*$ of the original MFG.

Under proper conditions, the Nash equilibrium $(\pi^*, \mathcal{L}^*)$ can be obtained via fixed-point updates, which generate a sequence $\{\pi_t, \mathcal{L}_t\}$ as follows. For any $t \geq 0$, in the $t$-th iteration, we solve the MDP induced by $\mathcal{L}_t$ and let $\pi_t$ be the optimal policy. Then we update the mean-field state by letting $\mathcal{L}_{t+1}$ be the mean-field state obtained by letting every agent follow $\pi_t$. Under appropriate assumptions, the mapping from $\mathcal{L}_t$ to $\mathcal{L}_{t+1}$ is a contraction and thus such an iterative algorithm converges to the unique fixed-point of such a contractive mapping, which corresponds to $\mathcal{L}^*$ (Guo et al., 2019). Based on the contractive property, various reinforcement learning methods are proposed to approximately implement the fixed-point updates and find the Nash equilibrium $(\pi^*, \mathcal{L}^*)$ (Guo et al., 2019; 2020; Anahtarci et al., 2019b;a; 2020). However, such an approach requires solving a standard reinforcement learning problem approximately within each iteration, which itself is solved by an iterative algorithm such as Q-learning (Watkins & Dayan, 1992; Mnih et al., 2015; Bellemare et al., 2017) or actor-critic methods (Konda & Tsitsiklis, 2000; Haarnoja et al., 2018; Schulman et al., 2015; 2017). As a result, this approach leads to a *double-loop* iterative algorithm for solving MFG. When the state space $\mathcal{S}$ is enormous, function approximation tools such as deep neural networks are equipped to represent the value and policy functions in the reinforcement learning algorithm, making solving each inner subproblem computationally demanding.

To obtain a computationally efficient algorithm for MFG, we consider the following question:

Can we design a single-loop reinforcement learning algorithm for solving MFG which updates the policy and mean-field state simultaneously in each iteration?

For such a question, we provide an affirmative answer by proposing a fictitious play (Brown, 1951) policy optimization algorithm, where we view the policy $\pi$ and mean-field state $\mathcal{L}$ as the two players and update them simultaneously in each iteration. Fictitious play is a general algorithm framework for solving games where each player first infers the opponent and then improves its own policy based on the inferred opponent information. When it comes to MFG, in each iteration, the policy player $\pi$ first infers the mean-field state implicitly by solving a policy evaluation problem associated with $\pi$ on the MDP induced by $\mathcal{L}$. Then the policy $\pi$ is updated via a proximal policy optimization (PPO) (Schulman et al., 2017) step with entropy regularization, which is adopted to ensure the uniqueness of the Nash equilibrium. Meanwhile, the mean-field state $\mathcal{L}$ obtains its update direction by solving how the mean-field state evolves when all the agents execute policy $\pi$ with their state distribution being $\mathcal{L}$. Then $\mathcal{L}$ is updated towards this direction with some stepsize. Such an algorithm is single-loop as the mean-field state $\mathcal{L}$ is updated immediately when $\pi$ is updated.

Furthermore, since $\mathcal{L}$ is a distribution over the state space $\mathcal{S}$, when $\mathcal{S}$ is continuous, $\mathcal{L}$ lies in an infinite-dimensional space, which makes it computationally challenging to be updated. To overcome this challenge, we employ a succinct representation of $\mathcal{L}$ via kernel mean embedding, which maps $\mathcal{L}$ to an element in a reproducing kernel Hilbert space (RKHS) (Smola et al., 2007; Gretton et al., 2008; Sriperumbudur et al., 2010). Such a mechanism enables us to update the mean-field state within RKHS, which can be computed efficiently.

When the stepsizes for policy and mean-field state updates are properly chosen, we prove that our single-loop fictitious play algorithm converges to the entropy-regularized Nash equilibrium at a sublinear $\widetilde{\mathcal{O}}(T^{-1/5})$-rate, where $T$ is the total number of iterations and $\widetilde{\mathcal{O}}(\cdot)$ hides logarithmic terms. To our best knowledge, we establish the first single-loop reinforcement learning algorithm for mean-field game with finite-time convergence guarantee to Nash equilibrium.

**Our Contributions.** Our contributions are two-fold. First, we propose a single-loop fictitious play algorithm that updates both the policy and the mean-field state simultaneously in each iteration, where the policy is updated via entropy-regularized proximal policy optimization. Moreover, we utilize kernel mean embedding to represent the mean-field states and the policy update subroutine

can readily incorporate any function approximation tools to represent both the value and policy functions, which makes our fictitious play method a general algorithmic framework that is able to handle MFG with continuous state space. Second, we prove that the policy and mean-field state sequence generated by the proposed algorithm converges to the Nash equilibrium of the MFG at a sublinear $\widetilde{\mathcal{O}}(T^{-1/5})$ rate.

**Related Works.** Our work belongs to the literature on discrete-time MFG. A variety of works have focused on the existence of a Nash equilibrium and the behavior of Nash equilibrium as the number of agents goes to infinity under various settings of MFG. See, e.g., Gomes et al. (2010); Tembine & Huang (2011); Moon & Başar (2014); Biswas (2015); Saldi et al. (2018b;a; 2019); Więcek (2020) and the references therein. In addition, our work is more related to the line of research that aims to solve MFG via reinforcement learning methods. Most of the existing works propose to find the Nash equilibrium via fixed-point iterations in space of the mean-field states, which requires solving an MDP induced by a mean-field state within each iteration (Guo et al., 2019; 2020; Anahtarci et al., 2019a;b; Fu et al., 2019; uz Zaman et al., 2020; Anahtarci et al., 2020). Among these works, Guo et al. (2019; 2020); Anahtarci et al. (2019a;b; 2020) propose to solve each MDP via Q-learning Watkins & Dayan (1992) or approximated value iteration (Munos & Szepesvári, 2008), whereas Fu et al. (2019); uz Zaman et al. (2020) solve each MDP using actor-critic (Konda & Tsitsiklis, 2000) under the linear-quadratic setting. Furthermore, more closely related works are Elie et al. (2019); Perrin et al. (2020), which study the convergence of a version of fictitious play for MFG. Similar to our algorithm, their fictitious play also regards the policy and the mean-field state as the two players. However, for policy update, they compute the best response policy to the current mean-field state by solving the MDP induced by the mean-field state to approximate optimality, and the obtained policy is added to the set of previous policy iterates to form a mixture policy. As a result, their algorithm is double-loop in essence due to solving an MDP in each iteration. In contrast, our fictitious play is single-loop — the policy is updated via a single PPO step in each iteration, and the mean-field state is updated before the policy solves any MDP associated with a mean-field state.

**Notations.** We use $\|\cdot\|_1$ to denote the vector $\ell_1$-norm, and $\Delta(\mathcal{D})$ the probability simplex over $\mathcal{D}$. The Kullback-Leibler (KL) divergence between $p_1, p_2 \in \Delta(\mathcal{A})$ is defined as $D_{\mathrm{KL}}(p_1 \| p_2) := \sum_{a \in \mathcal{A}} p_1(a) \log \frac{p_1(a)}{p_2(a)}$. Let $\mathbf{1}_n \in \mathbb{R}^n$ denote the all-one vector. For two quantities $x$ and $y$ that may depend on problem parameters ($|\mathcal{A}|, \gamma$, etc.), if $x \geq Cy$ holds for a universals constant $C > 0$, we write $x \gtrsim y$, $x = \Omega(y)$ and $y = \mathcal{O}(x)$. We use $\widetilde{O}(\cdot)$ to denote $\mathcal{O}(\cdot)$ ignoring logarithmic factors.

## 2 BACKGROUND AND PRELIMINARIES

In this section, we first review the standard setting of mean-field games (MFG) from Guo et al. (2019), and then introduce a more general MFG with mean embedding and entropy regularization.

### 2.1 MEAN-FIELD GAMES

Consider a discrete-time Markov game involving an infinite number of identical and interchangeable agents. Let $\mathcal{S} \subseteq \mathbb{R}^d$ and $\mathcal{A} \subseteq \mathbb{R}^p$ be the state space and action space, respectively, that are common to the agents. We assume that $\mathcal{S}$ is compact and $\mathcal{A}$ is finite. The reward and the state dynamic for each agent depend on the collective behavior of all agents through the mean-field state, i.e., the *distribution* of the states of all agents. As the agents are homogeneous and interchangeable, one can focus on a single agent representative of the population. Let $r : \mathcal{S} \times \mathcal{A} \times \Delta(\mathcal{S}) \to [0, R_{\max}]$ be the (bounded) reward function and $\mathrm{P}: \mathcal{S} \times \mathcal{A} \times \Delta(\mathcal{S}) \to \Delta(\mathcal{S})$ be the state transition kernel. At each time $t$, the representative agent is in state $s_t \in \mathcal{S}$, and the probability distribution of $s_t$, denoted by $\mathcal{L}_t \in \Delta(\mathcal{S})$, corresponds to the mean-field state. Upon taking an action $a_t \in \mathcal{A}$, the agent receives a reward $r(s_t, a_t, \mathcal{L}_t)$ and transitions to a new state $s_{t+1} \sim \mathrm{P}(\cdot|s_t, a_t, \mathcal{L}_t)$. A Markovian policy for the agent is a function $\pi : \mathcal{S} \to \Delta(\mathcal{A})$ that maps her own state to a distribution over actions,[1] i.e., $\pi(a|s)$ is the probability of taking action $a$ in state $s$. Let $\Pi$ be the set of all Markovian policies.

---

[1] In general, the policy may be a function of the mean-field state $\mathcal{L}_t$ as well. We have suppressed this dependency since our ultimate goal is to find a *stationary* equilibrium, under which the mean-field state remains fixed over time. See Guo et al. (2019); Saldi et al. (2018b) for a similar treatment.

When an agent is operating under a policy $\pi \in \Pi$ and the mean-field population flow is $\mathcal{L} := (\mathcal{L}_t)_{t \geq 0}$, we define the expected cumulative discounted reward (or value function) of this agent as

$$V^\pi(s, \mathcal{L}) := \mathbb{E}\big[\textstyle\sum_{t=0}^\infty \gamma^t r(s_t, a_t, \mathcal{L}_t) \mid s_0 = s\big],$$

where $a_t \sim \pi(\cdot|s_t)$, $s_{t+1} \sim \mathrm{P}(\cdot|s_t, a_t, \mathcal{L}_t)$, and $\gamma \in (0, 1)$ is the discount factor. The goal of this agent is to find a policy $\pi$ that maximizes $V^\pi(s, \mathcal{L})$ while interacting with the mean-field $\mathcal{L}$.

We are interested in finding a *stationary (time-independent) Nash Equilibrium* (NE) of the game, which is a policy-population pair $(\pi^*, \mathcal{L}^*) \in \Pi \times \Delta(\mathcal{S})$ satisfying the following two properties:

- (Agent rationality) $V^{\pi^*}(s, \mathcal{L}^*) \geq V^\pi(s, \mathcal{L}^*), \forall \pi \in \Pi, s \in \mathcal{S}$.
- (Population consistency) $\mathcal{L}_t = \mathcal{L}^*, \forall t$ under policy $\pi^*$ with initial mean-field state $\mathcal{L}_0 = \mathcal{L}^*$.

That is, $\pi^*$ is the optimal policy under the mean-field $\mathcal{L}^*$, and $\mathcal{L}^*$ remains fixed under $\pi^*$. We formalize the notion of NE in Section 2.3 after introducing a more general setting of MFG.

## 2.2 MEAN EMBEDDING OF MEAN-FIELD STATES

Note that the mean-field state $\mathcal{L}^*$ is a distribution over the states. When the state space is continuous, the NE $(\pi^*, \mathcal{L}^*)$ is an infinite dimensional object, posing challenges for learning the NE. To overcome this challenge, we make use of a succinct representation of the mean-field via mean embedding, which embeds the mean-field states into a reproducing kernel Hilbert space (RKHS) (Smola et al., 2007; Gretton et al., 2008; Sriperumbudur et al., 2010). Specifically, given a positive definite kernel $k : \mathcal{S} \times \mathcal{S} \to \mathbb{R}$, let $\mathcal{H}$ be the associated RKHS endowed with the inner product $\langle \cdot, \cdot \rangle_\mathcal{H}$ and norm $\|\cdot\|_\mathcal{H}$. For each $\mathcal{L} \in \Delta(\mathcal{S})$, its mean embedding $\mu_\mathcal{L} \in \mathcal{H}$ is defined as

$$\mu_\mathcal{L}(s) := \mathbb{E}_{x \sim \mathcal{L}}\left[k(x, s)\right], \quad \forall s \in \mathcal{S}.$$

Let $\mathcal{M} := \{\mu_\mathcal{L} : \mathcal{L} \in \Delta(\mathcal{S})\} \subseteq \mathcal{H}$ be the set of all possible mean embeddings. Note that when $k$ is the identity kernel, we have $\mu_\mathcal{L} = \mathcal{L}$ and $\mathcal{M} = \Delta(\mathcal{S})$. On the other hand, when $k$ is more structured (e.g., with a fast decaying eigen spectrum), $\mathcal{M}$ has significantly lower complexity than the set $\Delta(\mathcal{S})$ of raw mean-field states.

We assume that the MFG respects the mean embedding structure, in the sense that the reward $r : \mathcal{S} \times \mathcal{A} \times \mathcal{M} \to [0, R_{\max}]$ and transition kernel $\mathrm{P} : \mathcal{S} \times \mathcal{A} \times \mathcal{M} \to \Delta(\mathcal{S})$ (with a slight abuse of notation) depend on the mean-field state $\mathcal{L}$ through its mean embedding representation $\mu_\mathcal{L}$. In particular, at each time $t$ with state $s_t$ and mean-field state $\mathcal{L}_t$, the representative agent takes action $a_t \sim \pi(\cdot|s_t)$, receives reward $r(s_t, a_t, \mu_{\mathcal{L}_t})$ and then transitions to a new state $s_{t+1} \sim \mathrm{P}(\cdot|s_t, a_t, \mu_{\mathcal{L}_t})$. The NE of the game is defined analogously. As mentioned, when $k$ is the identity kernel, the above setting reduces to the standard setting in Section 2.1 with raw-mean field states.

We impose a standard regularity condition on the kernel $k$.

**Assumption 1.** *The kernel $k$ is bounded and universal, in the sense that $k(s, s) \leq 1, \forall s \in \mathcal{S}$ and the corresponding RKHS $\mathcal{H}$ is dense w.r.t. the $L_\infty$ norm in the space of continuous functions on $\mathcal{S}$.*

Assumption 1 is standard in the kernel learning literature (Caponnetto & De Vito, 2007; Muandet et al., 2012; Szabó et al., 2015; Lin et al., 2017). When the kernel is bounded, the embedding of each $\mathcal{L} \in \Delta(\mathcal{S})$ satisfies $\|\mu_\mathcal{L}\|_\mathcal{H} \leq \int_{x \sim \mathcal{L}} \|k(x, \cdot)\|_\mathcal{H} \, dx \leq 1$. When one uses a universal kernel (e.g., Gaussian or Laplace kernel), the mean embedding mapping is injective and hence each embedding $\mu \in \mathcal{M}$ uniquely characterizes a distribution $\mathcal{L}$ in $\Delta(\mathcal{S})$ (Gretton et al., 2008; 2012).

## 2.3 ENTROPY REGULARIZATION

To ensure the uniqueness of the NE and achieve fast algorithmic convergence, we use an entropy regularization approach (Cen et al., 2020; Shani et al., 2019; Nachum et al., 2017), which augments the standard expected reward objective with an entropy term of the policy. In particular, we define the entropy-regularized value function as

$$V_\mu^{\lambda, \pi}(s) := \mathbb{E}_{a_t \sim \pi(\cdot|s_t), s_{t+1} \sim \mathrm{P}(\cdot|s_t, a_t, \mu)}\left[\sum_{t=0}^\infty \gamma^t [r(s_t, a_t, \mu) - \lambda \log \pi(a_t|s_t)] \mid s_0 = s\right],$$

where the parameter $\lambda > 0$ controls the regularization level and $\mu$ is the mean-embedding of some given mean-field state (fixed over time). Equivalently, one may view $V_\mu^{\lambda, \pi}$ as the usual value function

of $\pi$ with an entropy-regularized reward

$$r_\mu^{\lambda,\pi}(s,a) := r(s,a,\mu) - \lambda \log \pi(a|s), \qquad \forall s \in \mathcal{S}, a \in \mathcal{A}. \tag{1}$$

Also define the $Q$-function of a policy $\pi$ as

$$Q_\mu^{\lambda,\pi}(s,a) = r(s,a,\mu) + \gamma \mathbb{E}\left[V_\mu^{\lambda,\pi}(s_1) \mid s_0 = s, a_0 = a\right], \tag{2}$$

which is related to the value function as

$$V_\mu^{\lambda,\pi}(s) = \mathbb{E}_{a\sim\pi(\cdot|s)}\left[Q_\mu^{\lambda,\pi}(s,a) - \lambda \log \pi(a|s)\right] = \left\langle Q_\mu^{\lambda,\pi}(s,\cdot), \pi(\cdot|s)\right\rangle + \mathbb{H}\left(\pi(\cdot|s)\right), \tag{3}$$

where $\mathbb{H}\left(\pi(\cdot|s)\right) := -\sum_a \pi(a|s)\log\pi(a|s)$ is the Shannon entropy of the distribution $\pi(\cdot|s)$. Since the reward function $r$ is assumed to be $R_{\max}$-bounded, it is easy to show that the Q-function is also bounded as $\left\|Q_\mu^{\lambda,\pi}\right\|_\infty \leq Q_{\max} := (R_{\max} + \gamma\lambda\log|\mathcal{A}|)/(1-\gamma)$; see Lemma 5.

**Single-Agent MDP.** When the mean-field state and its mean-embedding remain fixed over time, i.e., $\mathcal{L}_t = \mathcal{L}$ and $\mu_{\mathcal{L}_t} = \mu, \forall t$, a representative agent aims to solve the optimization problem

$$\max_{\pi:\mathcal{S}\to\Delta(\mathcal{A})} V_\mu^{\lambda,\pi}(s) \tag{4}$$

for each $s \in \mathcal{S}$. This problem corresponds to finding the (entropy-regularized) optimal policy for a single-agent discounted MDP, denoted by $\mathrm{MDP}_\mu := (\mathcal{S}, \mathcal{A}, \mathrm{P}(\cdot|\cdot,\cdot,\mu), r(\cdot,\cdot,\mu), \gamma)$, that is induced by $\mu \in \mathcal{M}$. Let $\pi_\mu^{\lambda,*}$ be the optimal solution to the problem (4), that is, the optimal regularized policy of $\mathrm{MDP}_\mu$. The optimal policy is unique whenever $\lambda > 0$. One can thus define a mapping $\Gamma_1^\lambda : \mathcal{M} \to \Pi$ via $\Gamma_1^\lambda(\mu) = \pi_\mu^{\lambda,*}$, which maps each embedded mean-field state $\mu$ to the optimal regularized policy $\pi_\mu^{\lambda,*}$ of $\mathrm{MDP}_\mu$. Let $Q_\mu^{\lambda,*}$ be the optimal regularized Q-function corresponding to the optimal policy $\pi_\mu^{\lambda,*}$.

Throughout the paper, we fix a state distribution $\nu_0 \in \Delta(\mathcal{S})$, which will serve as the initial state of our policy optimization algorithm. For each $\mu \in \mathcal{M}$ and a policy $\pi : \mathcal{S} \to \Delta(\mathcal{A})$, define

$$J_\mu^\lambda(\pi) := \mathbb{E}_{s\sim\nu_0}\left[V_\mu^{\lambda,\pi}(s)\right] \tag{5}$$

as the expectation of the value function $V_\mu^{\lambda,\pi}(s)$ of policy $\pi$ on the regularized $\mathrm{MDP}_\mu$. We define the discounted state visitation distribution $\rho_\mu^\pi$ induced by a policy $\pi$ on $\mathrm{MDP}_\mu$ as:

$$\rho_\mu^\pi(s) := (1-\gamma)\sum_{t=0}^{\infty}\gamma^t\mathbb{P}(s_t = s), \tag{6}$$

where $\mathbb{P}(s_t = s)$ is the state distribution when $s_0 \stackrel{\text{i.i.d.}}{\sim} \nu_0$ and the actions are chosen according to $\pi$.

**Mean-field Dynamics.** When all agents follow the same policy $\pi$, we can define another mapping $\Gamma_2 : \Pi \times \mathcal{M} \to \mathcal{M}$ that describes the dynamic of the embedded mean-field state. In particular, given the current embedding $\mu$ corresponding to some mean-field state $\mathcal{L}$, the next embedded mean-field state $\mu^+ = \Gamma_2(\pi,\mu)$ is given by

$$\mathcal{L}^+(s') = \int_\mathcal{S}\sum_{a\in\mathcal{A}}\mathcal{L}(s)\pi(a|s)\mathrm{P}(s'|s,a,\mu)\mathrm{d}s, \qquad \mu^+ = \mu_{\mathcal{L}^+}. \tag{7}$$

Note that the evolution of the mean-field depends on the agents' policy in a deterministic manner.

**Entropy-regularized Mean-field Nash Equilibrium (NE).** With the above notations, we can formally define our notion of equilibrium.

**Definition 1.** A stationary (time-independent) entropy-regularized Nash equilibrium for the MFG is a policy-population pair $(\pi^*, \mu^*) \in \Pi \times \mathcal{M}$ that satisfies

$$\begin{aligned}\text{(agent rationality)} \qquad & \pi^* = \Gamma_1^\lambda(\mu^*), \\ \text{(population consistency)} \qquad & \mu^* = \Gamma_2(\pi^*, \mu^*).\end{aligned}$$

When $\lambda = 0$, the above definition reduces to that of the (unregularized) NE discussed in Section 2.1, which requires $\pi^*$ to the unregularized optimal policy of $\mathrm{MDP}_{\mu^*}$. For general values of $\lambda$, the regularized NE $(\pi^*, \mu^*)$ approximates the unregularized NE (Geist et al., 2019), in the sense that $\pi^*$ is an approximate optimal policy of $\mathrm{MDP}_{\mu^*}$ satisfying

$$\max_{\pi\in\Pi}\{J_{\mu^*}^0(\pi)\} - J_{\mu^*}^\lambda(\pi^*) \leq \lambda\log|\mathcal{A}|/(1-\gamma).$$

One may further define the composite mapping $\Lambda^\lambda : \mathcal{M} \to \mathcal{M}$ as $\Lambda^\lambda(\mu) = \Gamma_2\left(\Gamma_1^\lambda(\mu), \mu\right)$. When $\Lambda^\lambda$ is a contraction, the regularized NE exists and is unique (Guo et al., 2019). Moreover, the iterates $\{(\pi_t, \mu_t)\}_{t \geq 0}$ given by the two-step update

$$\pi_t = \Gamma_1^\lambda(\mu_t), \qquad \mu_{t+1} = \Gamma_2(\pi_t, \mu_t)$$

converge to the regularized NE at a linear rate. Note that the first step above requires an oracle for computing the exact optimal policy $\pi_{\mu_t}^{\lambda,*}$. In most cases, such an exact oracle is not available; various single-agent reinforcement learning algorithms have been considered for computing an approximate optimal policy, including Q-learning (Guo et al., 2019) and policy gradient methods (Guo et al., 2020; Subramanian & Mahajan, 2019). The recent work by Elie et al. (2019) considers fictitious play iterative learning scheme. We remark that their convergence guarantee requires being able to compute the approximate optimal policy to an arbitrary precision with high probability.

## 3 FICTITIOUS PLAY ALGORITHM FOR MFG

In this section, we present a fictitious play algorithm, which simultaneously estimates the policy $\pi^*$ and the embedded mean-field state $\mu^*$ of the NE. As given in Algorithm 1, each iteration of the algorithm involves three steps: policy evaluation (line 3), policy improvement (line 4), and updating the embedded mean-field state (line 5). Below we explain each step in more details.

---

**Algorithm 1** Mean-Embedded Fictitious Play

1: Input: initial estimate $(\pi_0, \mu_0)$, step size sequence $\{\alpha_t, \beta_t\}_{t \geq 0}$, mixing parameter $\eta$.
2: **for** Iteration $t = 0, 1, 2, \ldots, T-1$ **do**
3:     (Policy evaluation step) Compute an approximate version $\widehat{Q}_t^\lambda : \mathcal{S} \times \mathcal{A} \to [0, Q_{\max}]$ of the Q-function $Q_{\mu_t}^{\lambda,\pi_t}$ of policy $\pi_t$ with respect to the entropy-regularized $\text{MDP}_{\mu_t}$
4:     (Policy improvement step) Update the policy by

$$\widehat{\pi}_{t+1}(\cdot|s) \propto (\pi_t(\cdot|s))^{1-\alpha_t\lambda} \exp\left(\alpha_t \widehat{Q}_t^\lambda(s, \cdot)\right) \qquad (8)$$

$$\pi_{t+1}(\cdot|s) = (1-\eta)\widehat{\pi}_{t+1}(\cdot|s) + \eta \cdot \mathbf{1}_{|\mathcal{A}|}(\cdot)/|\mathcal{A}| \qquad (9)$$

5:     Update the embedded mean-field state by

$$\mu_{t+1} = (1-\beta_t)\mu_t + \beta_t \cdot \Gamma_2(\pi_{t+1}, \mu_t). \qquad (10)$$

6: **end for**
7: Output: $\left\{(\pi_t, \mu_t)\right\}_{t=1,\ldots,T}$

---

**Policy Evaluation.** In each iteration, we first evaluate the current policy $\pi_t$ with respect to the regularized single-agent $\text{MDP}_{\mu_t}$ induced by the current mean-field estimate $\mu_t$. In particular, we compute an approximation $\widehat{Q}_t^\lambda$ of the true Q-function $Q_t^\lambda := Q_{\mu_t}^{\lambda,\pi_t}$, which can be done using, e.g., TD(0) or LSTD methods. Our theorem characterizes how convergence depends on the policy evaluation error in this step.

**Policy Improvement.** To update our policy estimate $\pi_t$, we first compute an intermediate policy $\widehat{\pi}_{t+1}$ by a *single* policy improvement step: for each $s \in \mathcal{S}$,

$$\widehat{\pi}_{t+1}(\cdot|s) = \underset{\pi(\cdot|s) \in \Delta(\mathcal{A})}{\operatorname{argmax}} \left\{\alpha_t\langle \widehat{Q}_t^\lambda(s, \cdot) - \lambda\log\pi_t(\cdot|s), \pi(\cdot|s) - \pi_t(\cdot|s)\rangle - D_{\text{KL}}\left(\pi(\cdot|s)\,\|\,\pi_t(\cdot|s)\right)\right\},$$
$$(11)$$

where $\alpha_t > 0$ is the stepsize. This step corresponds to one iteration of Proximal Policy Optimization (PPO) (Schulman et al., 2017). It can also be viewed as one mirror descent iteration, where the shifted Q-function $\widehat{Q}_t^\lambda(s, \cdot) - \lambda\log\pi_t(\cdot|s)$ plays the role of the gradient. The maximizer $\widehat{\pi}_{t+1}$ in equation (11) can be computed in closed form as done in equation (8) in Algorithm 1. We then compute the new policy $\pi_{t+1}$ by mixing $\widehat{\pi}_{t+1}$ with a small amount of uniform distribution, as done in equation (9). "Mixing in" a uniform distribution is a standard technique to prevent the policy from approaching the boundary of the probability simplex and becoming degenerate. Doing so allows us to upper bound a quantity of the form $D_{\text{KL}}\left(p\,\|\,\pi_{t+1}(\cdot|s)\right)$ (cf. Lemma 2), which otherwise may be infinite. It also ensures that the KL divergence satisfies a Lipschitz condition (cf. Lemma 3).

**Mean-field Update.** We next compute an updated (embedded) mean-field state $\mu_{t+1}$ as a weighted average of the current $\mu_t$ and the mean-field state $\Gamma_2(\pi_{t+1}, \mu_t)$ induced by the new policy $\pi_{t+1}$, namely, $\mu_{t+1} = (1 - \beta_t)\mu_t + \beta_t \cdot \Gamma_2(\pi_{t+1}, \mu_t)$, where $\beta_t \in (0, 1)$ is the stepsize. This update can be viewed as a single step of the (soft) fixed point iteration for the equation $\mu = \Gamma_2(\pi_{t+1}, \mu)$.

We remark that our algorithm is similar to the classical fictitious play approach for finding NEs, where each agent plays a response to the empirical average of its opponent's past behaviors. In our algorithm, the representative agent views the population of all agents collectively as an opponent. Expanding the recursion (8) and ignoring the difference between $\widehat{\pi}_{t+1}$ and $\pi_{t+1}$, we can write the policy $\pi_{t+1}$ as

$$\pi_{t+1}(\cdot|s) \propto \exp\left(\sum_{\tau=0}^{t} w_\tau \widehat{Q}_\tau^\lambda(s, \cdot)\right)$$

for some positive weights $\{w_\tau\}$. Therefore, the representative agent is playing a policy that responds to the (weighted) average of all previous Q functions, which reflects the representative agent's belief on the aggregate population policy.

Also note that our algorithm only performs a single policy improvement step to compute the updated policy $\pi_{t+1}$. It is unnecessary to compute the exact optimal policy $\pi_{t+1}^* = \Gamma_1^\lambda(\mu_t)$ under $\mu_t$ (which would require an inner loop for solving $\text{MDP}_{\mu_t}$), as $\mu_t$ is only an approximate anyway of the true NE mean-field $\mu^*$. Our algorithm updates $\pi_t$ and $\mu_t$ simultaneously within a single loop.

## 4 MAIN RESULTS

In this section, we establish the theoretical guarantees on learning the regularized NE $(\pi^*, \mu^*)$ of the MFG for our fictitious play algorithm. To state our theorem, we first discuss several regularity assumptions on the MFG model. Recall the definition (6) of the discounted state visitation distribution and let $\rho^* := \rho_{\mu^*}^{\pi^*} \in \Delta(\mathcal{S})$ be the visitation distribution induced by the NE $(\pi^*, \mu^*)$. We make use of the following distance metric between two policies $\pi, \pi' \in \Pi$:

$$D(\pi, \pi') := \mathbb{E}_{s \sim \rho^*}\left[\|\pi(\cdot|s) - \pi'(\cdot|s)\|_1\right]. \tag{12}$$

As in the classical MFG literature (Guo et al., 2020; Saldi et al., 2018b), we assume certain Lipschitz properties for the two mappings $\Gamma_1^\lambda : \mathcal{M} \to \Pi$ and $\Gamma_2 : \Pi \times \mathcal{M} \to \mathcal{M}$ defined in Section 2.3. The first assumption states that $\Gamma_1^\lambda(\mu)$ is Lipschitz in the mean-embedded mean-field state $\mu$ with respect to the RKHS norm.

**Assumption 2.** *There exists a constant $d_1 > 0$, such that for any $\mu, \mu' \in \mathcal{M}$, it holds that*

$$D\left(\Gamma_1^\lambda(\mu), \Gamma_1^\lambda(\mu')\right) \leq d_1 \|\mu - \mu'\|_{\mathcal{H}}.$$

The second assumption states that $\Gamma_2(\pi, \mu)$ is Lipschitz in each of its arguments when the other argument is fixed.

**Assumption 3.** *There exist constants $d_2 > 0, d_3 > 0$ such that for any policies $\pi, \pi' \in \Pi$ and embedded mean-field states $\mu, \mu' \in \mathcal{M}$, it holds that*

$$\|\Gamma_2(\pi, \mu) - \Gamma_2(\pi', \mu)\|_{\mathcal{H}} \leq d_2 D\left(\pi, \pi'\right), \quad \|\Gamma_2(\pi, \mu) - \Gamma_2(\pi, \mu')\|_{\mathcal{H}} \leq d_3 \|\mu - \mu'\|_{\mathcal{H}}.$$

Assumptions 2 and 3 immediately imply Lipschitzness of the composite mapping $\Lambda^\lambda : \mathcal{M} \to \mathcal{M}$, which we recall is defined as $\Lambda^\lambda(\mu) = \Gamma_2\left(\Gamma_1^\lambda(\mu), \mu\right)$. The proof is provided in Appendix D.1.

**Lemma 1.** *Suppose Assumptions 2 and 3 hold. Then for each $\mu, \mu' \in \mathcal{M}$, it holds that*

$$\left\|\Lambda^\lambda(\mu) - \Lambda^\lambda(\mu')\right\|_{\mathcal{H}} \leq (d_1 d_2 + d_3) \|\mu - \mu'\|_{\mathcal{H}}.$$

We next impose an assumption on the boundedness of certain concentrability coefficients. This type of assumption, standard in analysis of policy optimization algorithms (Kakade & Langford, 2002; Shani et al., 2019; Bhandari & Russo, 2019; Agarwal et al., 2020), allows one to define the policy optimization error in an average-case sense with respect to appropriate distributions over the states.

**Assumption 4** (Finite Concentrability Coefficients). *There exist two constants $C_\rho, \overline{C}_\rho > 0$ such that for each $\mu \in \mathcal{M}$, it holds that*

$$\left\|\rho_\mu^{\pi_\mu^{\lambda,*}}/\rho^*\right\|_\infty := \sup_s \left[\rho_\mu^{\pi_\mu^{\lambda,*}}(s)/\rho^*(s)\right] \leq C_\rho \quad and \quad \left\{\mathbb{E}_{s \sim \rho_\mu^{\pi_\mu^{\lambda,*}}}\left[\left|\rho^*(s)/\rho_\mu^{\pi_\mu^{\lambda,*}}(s)\right|^2\right]\right\}^{1/2} \leq \overline{C}_\rho.$$

Finally, our last assumption stipulates that the state visitation distributions are smooth with respect to the (embedded) mean-field states of the MFG. This assumption is analogous to those in the literature on MDP and two-player games (Fei et al., 2020; Radanovic et al., 2019), which requires the visitation distributions to be smooth with respect to the policy.

**Assumption 5.** *There exists a constant $d_0 > 0$, such that for any $\mu, \mu' \in \mathcal{M}$, it holds that*

$$\left\| \rho_\mu^{\pi_\mu^{\lambda,*}} - \rho_{\mu'}^{\pi_{\mu'}^{\lambda,*}} \right\|_1 \leq d_0 \left\| \mu - \mu' \right\|_{\mathcal{H}}.$$

We now state our theoretical guarantees on the convergences of the policy-population sequence $\{\pi_t, \mu_t\}$ in Algorithm 1 to the NE $\{\pi^*, \mu^*\}$. For the estimates of the embedded mean-field states, it is natural to consider the distance $\|\mu_t - \mu^*\|_{\mathcal{H}}$ in RKHS norm. For convergence to NE policy $\mu^*$, recall that $\mu^*$ is the optimal policy to $\mathrm{MDP}_{\mu^*}$, and each iteration of our algorithm involves a single policy improvement step to compute $\pi_{t+1}$ rather than solving $\mathrm{MDP}_{\mu_t}$ to its optimal policy $\pi_{t+1}^* := \Gamma_1^\lambda(\mu_t)$. As such, we analyze the difference between these two policies in terms of $D\left(\pi_{t+1}, \pi_{t+1}^*\right)$, where the metric $D$ is defined in equation (12). Also let $\rho_t^* := \rho_{\mu_t}^{\pi_{t+1}^*}$ denote the discounted visitation distribution induced by the optimal policy $\pi_{t+1}^*$ of $\mathrm{MDP}_{\mu_t}$.[2] With the above considerations in mind, we have the following theorem, which is proved in Appendix B. .

**Theorem 1.** *Suppose that Assumptions 1–5 hold and $d_1 d_2 + d_3 < 1$ and that the error in the policy evaluation step in Algorithm 1 satisfies*

$$\mathbb{E}_{s \sim \rho_t^*}\left[ \left\| Q_t^\lambda(s, \cdot) - \widehat{Q}_t^\lambda(s, \cdot) \right\|_\infty^2 \right] \leq \varepsilon^2, \qquad \forall t \in [T].$$

*With the choice of*

$$\eta = c_\eta T^{-1}, \qquad \alpha_t \equiv \alpha = c_\alpha T^{-2/5}, \qquad \beta_t \equiv \beta = c_\beta T^{-4/5},$$

*for some universal constants $c_\eta > 0$, $c_\alpha > 0$ and $c_\beta > 0$ in Algorithm 1, the resulting policy and embedded mean-field state sequence $\{(\pi_t, \mu_t)\}_{t=1}^T$ satisfy*

$$D\left(\frac{1}{T}\sum_{t=1}^T \pi_t, \frac{1}{T}\sum_{t=1}^T \pi_t^*\right) \leq \frac{1}{T}\sum_{t=1}^T D(\pi_t, \pi_t^*) \lesssim \frac{1}{\sqrt{\lambda}} \cdot \left( \sqrt{\log T} \cdot T^{-1/5} + \sqrt{\varepsilon} \right), \qquad (13)$$

$$\left\| \frac{1}{T}\sum_{t=1}^T \mu_t - \mu^* \right\|_{\mathcal{H}} \leq \frac{1}{T}\sum_{t=1}^T \|\mu_t - \mu^*\|_{\mathcal{H}} \lesssim \frac{1}{\sqrt{\lambda}} \cdot \left( \sqrt{\log T} \cdot T^{-1/5} + \sqrt{\varepsilon} \right). \qquad (14)$$

Theorem 1 bounds the distance between $\pi_t$ and the optimal policy $\pi_t^*$ of $\mathrm{MDP}_{\mu_t^*}$. By directly measuring the distance between $\pi_t$ and the NE policy $\pi^*$, we can define the notion of an $\delta$-approximate NE of the game.

**Definition 2.** For each $\delta > 0$, a policy-population pair $(\pi, \mu)$ is called an $\delta$-approximate (entropy-regularized) NE of the MFG if

$$D(\pi, \pi^*) \leq \delta \quad \text{and} \quad \|\mu - \mu^*\|_{\mathcal{H}} \leq \delta.$$

The following corollary of Theorem 1 shows that after $T$ iterations of our algorithm, the average policy-population pair $\left(\frac{1}{T}\sum_{t=1}^T \pi_t, \frac{1}{T}\sum_{t=1}^T \mu_t\right)$ is an $\widetilde{\mathcal{O}}\left(T^{-1/5}\right)$-approximate NE.

**Corollary 1.** *Under the assumptions of Theorem 1, we have*

$$D\left(\frac{1}{T}\sum_{t=1}^T \pi_t, \pi^*\right) + \left\| \frac{1}{T}\sum_{t=1}^T \mu_t - \mu^* \right\|_{\mathcal{H}} \lesssim \frac{1}{\sqrt{\lambda}} \cdot \left( \sqrt{\log T} \cdot T^{-1/5} + \sqrt{\varepsilon} \right).$$

We prove this corollary in Appendix C.

The above results require an $\ell_2$-error of $\varepsilon$ for policy evaluation. A variety of algorithms have been shown to achieve such a guarantees, including TD(0) and LSTD (Bhandari et al., 2018).

We also remark that the $\ell_\infty$ condition on concentrability coefficient in Assumption 4 can be relaxed to an $\ell_2$ condition of the form $\left\{ \mathbb{E}\left[ \left| \rho_\mu^{\pi_\mu^{\lambda,*}}(s)/\rho^*(s) \right|^2 \right] \right\}^{1/2} \leq C_\rho$, under which we can establish an $\widetilde{O}(T^{-1/9})$ convergence rate; see Theorem 2 and Corollary 2 in Appendix E for the details.

---

[2]The subscript in $\rho_t^*$ emphasizes that $\rho_t^*$ only depends on the mean-field state $\mu_t$ at time $t$ through $\pi_{t+1}^* = \Gamma_1^\lambda(\mu_t)$.

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

# Appendices

## A  TECHNICAL LEMMAS

**Lemma 2.** *Let $p^*$ and $p \in \Delta(\mathcal{A})$ and $\widehat{p} = (1 - \eta)p + \eta \frac{\mathbf{1}_{|\mathcal{A}|}}{|\mathcal{A}|}$. Then*

$$D_{\mathrm{KL}}\left(p^* \| \widehat{p}\right) \leq \log \frac{|\mathcal{A}|}{\eta},$$

$$D_{\mathrm{KL}}\left(p^* \| \widehat{p}\right) - D_{\mathrm{KL}}\left(p^* \| p\right) \leq 2\eta.$$

*Proof.* By definition we have

$$
\begin{aligned}
D_{\mathrm{KL}}\left(p^* \| \widehat{p}\right) &= \sum_{a \in \mathcal{A}} p^*(a) \log \frac{p^*(a)}{\widehat{p}(a)} \\
&= \sum_{a \in \mathcal{A}} p^*(a) \log \frac{p^*(a)}{(1 - \eta)p(a) + \frac{\eta}{|\mathcal{A}|}} \\
&\leq \sum_{a \in \mathcal{A}} p^*(a) \log \frac{1}{0 + \frac{\eta}{|\mathcal{A}|}} \\
&= \log \frac{|\mathcal{A}|}{\eta},
\end{aligned}
$$

thereby proving the first inequality.

Note that

$$D_{\mathrm{KL}}\left(p^* \| \widehat{p}\right) - D_{\mathrm{KL}}\left(p^* \| p\right) = \sum_{a \in \mathcal{A}} p^*(a) \log \left( \frac{p(a)}{\widehat{p}(a)} \right). \tag{15}$$

If $\frac{p(a)}{\widehat{p}(a)} \leq 1$ for all $a \in \mathcal{A}$ then we have

$$D_{\mathrm{KL}}\left(p^* \| \widehat{p}\right) - D_{\mathrm{KL}}\left(p^* \| p\right) \leq 0;$$

otherwise, there exists $a'$ such that $p(a') \geq \widehat{p}(a')$ and we have

$$
\begin{aligned}
\log \left( \frac{p(a')}{\widehat{p}(a')} \right) &= \log \left( \frac{p(a')}{(1 - \eta)p(a') + \eta/|\mathcal{A}|} \right) \\
&\leq \log \left( \frac{p(a')}{(1 - \eta)p(a')} \right) \\
&\leq \frac{\eta}{1 - \eta} \leq 2\eta,
\end{aligned}
$$

where the third step follows from the fact that $\log(z) \leq z - 1$ for all $z > 0$ and the last step holds as $\eta \in [0, \frac{1}{2}]$. Therefore, we have $\log \left( \frac{p(a')}{\widehat{p}(a')} \right) \leq 2\eta$. Applying Holder's inequality to (15) completes the proof. $\qquad\square$

**Lemma 3.** *Let $x, y$ and $z \in \Delta(\mathcal{A})$. If $x(a) \geq \alpha_1$, $y(a) \geq \alpha_1$ and $z(a) \geq \alpha_2$ for all $a \in \mathcal{A}$, then*

$$D_{\mathrm{KL}}(x \| z) - D_{\mathrm{KL}}(y \| z) \leq \left( 1 + \log \frac{1}{\min\{\alpha_1, \alpha_2\}} \right) \cdot \|x - y\|_1.$$

*Proof.* Under the lower bound assumption of the lemma, we have

$$\frac{\mathrm{d}D_{\mathrm{KL}}(x \| z)}{\mathrm{d}x(a)} = 1 + \log \frac{x(a)}{z(a)} \leq 1 + \log \frac{1}{\alpha_2}$$

and

$$-\frac{\mathrm{d}D_{\mathrm{KL}}(x \| z)}{\mathrm{d}x(a)} \leq -1 - \log \alpha_1.$$

It follows that

$$\left\| \frac{\mathrm{d} D_{\mathrm{KL}}(x\|z)}{\mathrm{d}x(a)} \right\|_\infty \le \max\left\{ 1 + \log\frac{1}{\alpha_2}, -1 - \log\alpha_1 \right\} \le 1 + \log\frac{1}{\min\{\alpha_1, \alpha_2\}}.$$

Hence the function $x \mapsto D_{\mathrm{KL}}(x\|z)$ is Lipschitz w.r.t. $\|\cdot\|_1$, the dual norm of $\|\cdot\|_\infty$. $\qquad\square$

## B   PROOF OF THEOREM 1

In order to obtain an upper bound on the optimality gap

$$\sigma_\mu^t := \|\mu_t - \mu^*\|_{\mathcal{H}}, \tag{16}$$

where $\mu^*$ is the embedded mean-field state of the entropy regularized NE, we also need to estimate the gap between $\pi_{t+1}$ and the optimal solution to the entropy regularized $\mathrm{MDP}_{\mu_t}$. We define

$$\sigma_\pi^{t+1} := \mathbb{E}_{s\sim\rho_t^*}\left[ D_{\mathrm{KL}}\left( \pi_{t+1}^*(\cdot|s)\|\pi_{t+1}(\cdot|s) \right) \right] \tag{17}$$

to quantify the convergence of policy sequence.

Before proceeding, we establish the following properties of entropy regularized MDPs, which are central to the convergence analysis.

**Properties of Regularized MDP.**   The following lemma quantifies the performance difference between two policies for a regularized MDP — measured in terms of the expected total reward — through the Q-function and their KL-divergence. The proof is provided in Appendix D.2.

**Lemma 4** (Performance Difference). *For each $\mu \in \mathcal{M}$ and policies $\pi : \mathcal{S} \to \Delta(\mathcal{A})$, it holds that*

$$J_\mu^\lambda(\pi') - J_\mu^\lambda(\pi) + \frac{\lambda}{1-\gamma}\mathbb{E}_{s\sim\rho_\mu^{\pi'}}\left[ D_{\mathrm{KL}}\left( \pi'(\cdot|s)\|\pi(\cdot|s) \right) \right]$$

$$= \frac{1}{1-\gamma}\mathbb{E}_{s\sim\rho_\mu^{\pi'}}\left[ \langle Q_\mu^{\lambda,\pi}(s,\cdot) - \lambda\log\pi(\cdot|s), \pi'(\cdot|s) - \pi(\cdot|s) \rangle \right], \tag{18}$$

*where $\rho_\mu^{\pi'}$ is the discounted state visitation distribution induced by the policy $\pi'$ on $\mathrm{MDP}_\mu$.*

We can characterize the optimal policy $\pi_\mu^{\lambda,*}$ in terms of the optimal Q-function $Q_\mu^{\lambda,*}$ as a Boltzmann distribution of the form Cen et al. (2020); Nachum et al. (2017)

$$\pi_\mu^{\lambda,*}(a|s) \propto \exp\left( \frac{Q_\mu^{\lambda,*}(s,a)}{\lambda} \right). \tag{19}$$

For the setting where the reward function is bounded, we then can obtain a lower bound on $\pi_\mu^{\lambda,*}$, as stated in the following lemma. The proof is provided in Appendix D.3

**Lemma 5.** *Suppose that there exists a constant $R_{\max} > 0$ such that $0 \le \sup_{(s,a,\mu)\in\mathcal{S}\times\mathcal{A}\times\mathcal{M}} r(s,a,\mu) \le R_{\max}$. For each $\mu \in \mathcal{M}$, and each policy $\pi : \mathcal{S} \to \Delta(\mathcal{A})$, we have*

$$\left\| Q_\mu^{\lambda,\pi} \right\|_\infty \le Q_{\max} := \frac{R_{\max} + \gamma\lambda\log|\mathcal{A}|}{1-\gamma}.$$

*Also, the optimal policy $\pi_\mu^{\lambda,*}$ for the regularized $\mathrm{MDP}_\mu$ satisfies*

$$\pi_\mu^{\lambda,*}(a|s) \ge \frac{1}{e^{Q_{\max}/\lambda}|\mathcal{A}|}, \forall s \in \mathcal{S}, a \in \mathcal{A}.$$

**Convergence Analysis.**   We now move to the convergence analysis. For clarity of exposition, we use $\mathbb{E}_\rho\left[ \|\pi - \pi'\|_1 \right]$ as shorthand for $\mathbb{E}_{s\sim\rho}\left[ \|\pi(\cdot|s) - \pi'(\cdot|s)\|_1 \right]$, where $\rho \in \Delta(\mathcal{S})$; we also use $\mathbb{E}_\rho\left[ D_{\mathrm{KL}}\left( \pi\|\pi' \right) \right]$ as shorthand for $\mathbb{E}_{s\sim\rho}\left[ D_{\mathrm{KL}}\left( \pi(\cdot|s)\|\pi'(\cdot|s) \right) \right]$. We recall that the step sizes are chosen as

$$\alpha_t \equiv \alpha = c_\alpha T^{-2/5}, \qquad \beta_t \equiv \beta = c_\beta T^{-4/5},$$

where the parameters $c_\alpha$ and $c_\beta$ satisfy that:

$$c_\alpha T^{-2/5}\lambda < 1, \qquad c_\beta T^{-4/5}\overline{d} < 1. \tag{20}$$

Here $\overline{d} := 1 - d_1 d_2 - d_3 > 0$, where $d_1$ appears in Assumption 2, and $d_2, d_3$ appear in Assumption 3.

**Step 1: Convergence of Policy.** To analyze the convergence of the optimality gap $\sigma_\mu^{t+1} = \|\mu_{t+1} - \mu^*\|_{\mathcal{H}}$, we first characterize the convergence behavior of the policy sequence $\{\pi_t\}_{t \geq 0}$. In particular, we establish a recursive relationship between $\sigma_\pi^{t+1} = \mathbb{E}_{s \sim \rho_t^*}\left[D_{\mathrm{KL}}\left(\pi_{t+1}^*(\cdot|s)\|\pi_{t+1}(\cdot|s)\right)\right]$ and $\sigma_\pi^t$, as stated in the following lemma. The proof is provided in Section B.1.

**Lemma 6.** *Under the setting of Theorem 1, for each $t \geq 1$, we have*

$$\sigma_\pi^{t+1} \leq (1 - \lambda\alpha_t)\sigma_\pi^t + (1 - \lambda\alpha_t)\left(d_0 \log \frac{|\mathcal{A}|}{\eta} + \kappa C_\rho d_1\right)\|\mu_{t-1} - \mu_t\|_{\mathcal{H}} + 2\varepsilon\alpha_t + \frac{Q_{\max}^2}{2}\alpha_t^2 + 2\eta, \tag{21}$$

*where $\kappa = \frac{4}{1-\gamma} \log \frac{|\mathcal{A}|}{\eta} + \frac{2R_{\max}}{\lambda(1-\gamma)}$.*

Recall that $\mu_t = (1 - \beta_{t-1})\mu_{t-1} + \beta_{t-1} \cdot \Gamma_2(\pi_t, \mu_{t-1})$. Under Assumption 1, we have

$$\|\mu_{t-1} - \mu_t\|_{\mathcal{H}} = \beta_{t-1}\|\mu_{t-1} - \Gamma_2(\pi_t, \mu_{t-1})\|_{\mathcal{H}} \leq 2\beta_{t-1}. \tag{22}$$

Lemma 6 implies that

$$\sigma_\pi^{t+1} \leq (1 - \lambda\alpha_t)\sigma_\pi^t + (1 - \lambda\alpha_t)\overline{C}_1\beta_{t-1} + 2\varepsilon\alpha_t + \frac{Q_{\max}^2}{2}\alpha_t^2 + 2\eta, \tag{23}$$

where we define

$$\overline{C}_1 := 2\left(d_0 \log \frac{|\mathcal{A}|}{\eta} + \kappa C_\rho d_1\right).$$

With $\alpha_t \equiv \alpha$, $\beta_t \equiv \beta$, from Equation (23) we have that

$$\sigma_\pi^t \leq \frac{1}{\lambda\alpha}\left(\sigma_\pi^t - \sigma_\pi^{t+1}\right) + \left(\frac{1}{\lambda\alpha} - 1\right)\overline{C}_1\beta + \frac{2\varepsilon}{\lambda} + \frac{Q_{\max}^2}{2\lambda}\alpha + \frac{2\eta}{\lambda\alpha}. \tag{24}$$

Summing over $\ell = 0, 2, \ldots T - 1$ on both sides of (24) and dividing by $t$ gives

$$\frac{1}{T}\sum_{t=0}^{T-1}\sigma_\pi^t \leq \frac{1}{T\lambda\alpha}\left(\sigma_\pi^0 - \sigma_\pi^T\right) + \left(\frac{1}{\lambda\alpha} - 1\right)\overline{C}_1\beta + \frac{2\varepsilon}{\lambda} + \frac{Q_{\max}^2}{2\lambda}\alpha + \frac{2\eta}{\lambda\alpha}$$

$$\leq \frac{1}{T\lambda\alpha}\sigma_\pi^0 + \frac{\overline{C}_1\beta}{\lambda\alpha} + \frac{2\varepsilon}{\lambda} + \frac{Q_{\max}^2}{2\lambda}\alpha + \frac{2\eta}{\lambda\alpha}. \tag{25}$$

When choosing $\alpha = \mathcal{O}(T^{-2/5})$, $\beta = \mathcal{O}(T^{-4/5})$ and $\eta = \mathcal{O}(T^{-1})$, we have $\overline{C}_1 = \mathcal{O}(\log T)$. Therefore, we obtain

$$\frac{1}{T}\sum_{t=0}^{T-1}\sigma_\pi^t \lesssim \frac{\log T}{\lambda T^{2/5}} + \frac{2\varepsilon}{\lambda}. \tag{26}$$

If we let $\mathsf{T}$ be a random number sampled uniformly from $\{1, \ldots, T\}$, then the above equation can be written equivalently as

$$\mathbb{E}_{\mathsf{T}}\left[\sigma_\pi^{\mathsf{T}}\right] \lesssim \frac{\log T}{\lambda T^{2/5}} + \frac{2\varepsilon}{\lambda}. \tag{27}$$

**Step 2: Convergence of Mean-field Embedding.** We now proceed to characterize the optimality gap for the embedded mean-field state. We obtain the following upper bound on the optimality gap $\sigma_\mu^{t+1} = \|\mu_{t+1} - \mu^*\|_{\mathcal{H}}$. The proof is provided in Section B.2.

**Lemma 7.** *Under the setting of Theorem 1, for each $t \geq 0$, we have*

$$\sigma_\mu^{t+1} \leq \left(1 - \beta_t\overline{d}\right)\sigma_\mu^t + d_2\overline{C}_\rho\beta_t\sqrt{\sigma_\pi^{t+1}},$$

*where $\overline{d} = 1 - d_1 d_2 - d_3 > 0$.*

Lemma 7 implies that

$$\sigma_\mu^t \leq \frac{1}{\overline{d}\beta_t}\left(\sigma_\mu^t - \sigma_\mu^{t+1}\right) + \frac{d_2\overline{C}_\rho}{\overline{d}}\sqrt{\sigma_\pi^{t+1}}. \tag{28}$$

With $\beta_t \equiv \beta = \mathcal{O}(T^{-4/5})$, averaging equation (28) over iteration $t = 0, \dots, T-1$, we obtain

$$
\frac{1}{T} \sum_{t=0}^{T-1} \sigma_\mu^t \leq \frac{1}{\overline{d}\beta T} \left( \sigma_\mu^0 - \sigma_\mu^T \right) + \frac{d_2 \overline{C}_\rho}{\overline{d}T} \sum_{t=0}^{T-1} \sqrt{\sigma_\pi^{t+1}}
$$

$$
\leq \frac{\sigma_\mu^0}{\overline{d}\beta T} + \frac{d_2 \overline{C}_\rho}{\overline{d}T} \sum_{t=0}^{T-1} \sqrt{\sigma_\pi^{t+1}}
$$

$$
\leq \frac{\sigma_\mu^0}{\overline{d}\beta T} + \frac{d_2 \overline{C}_\rho}{\overline{d}} \sqrt{\frac{1}{T} \sum_{t=0}^{T-1} \sigma_\pi^{t+1}},
$$

where the last inequality follows from Cauchy-Schwarz inequality.

From Eq. (26), we have

$$
\frac{1}{T} \sum_{t=0}^{T-1} \sigma_\mu^t \lesssim \frac{\sigma_\mu^0}{\overline{d}} T^{-1/5} + \frac{d_2 \overline{C}_\rho}{\overline{d}} \sqrt{\frac{\log T}{\lambda T^{2/5}} + \frac{2\varepsilon}{\lambda}}
$$

$$
\lesssim \sqrt{\frac{\log T}{\lambda T^{2/5}} + \frac{2\varepsilon}{\lambda}}
$$

$$
\lesssim \frac{1}{\sqrt{\lambda}} \left( \frac{\sqrt{\log T}}{T^{1/5}} + \sqrt{\varepsilon} \right).
$$

This equation, together with Jensen's inequality, proves equation (14) in Theorem 1.

Turning to equation (13) in Theorem 1, we have

$$
\frac{1}{T} \sum_{t=1}^{T} D\left( \pi_t, \pi_t^* \right) = \mathbb{E}_\mathsf{T} \left[ D\left( \pi_\mathsf{T}, \pi_\mathsf{T}^* \right) \right]
$$

$$
= \mathbb{E}_\mathsf{T} \mathbb{E}_{s \sim \rho^*} \left[ \left\| \pi_\mathsf{T}^*(\cdot|s) - \pi_\mathsf{T}(\cdot|s) \right\|_1 \right]
$$

$$
= \mathbb{E}_\mathsf{T} \mathbb{E}_{s \sim \rho_{\mathsf{T}-1}^*} \left[ \frac{\rho^*(s)}{\rho_{\mathsf{T}-1}^*(s)} \left\| \pi_\mathsf{T}^*(\cdot|s) - \pi_\mathsf{T}(\cdot|s) \right\|_1 \right]
$$

$$
\overset{(i)}{\leq} \sqrt{ \mathbb{E}_\mathsf{T} \mathbb{E}_{s \sim \rho_{\mathsf{T}-1}^*} \left[ \left| \frac{\rho^*(s)}{\rho_{\mathsf{T}-1}^*(s)} \right|^2 \right] \cdot \mathbb{E}_\mathsf{T} \mathbb{E}_{s \sim \rho_{\mathsf{T}-1}^*} \left[ \left\| \pi_\mathsf{T}^*(\cdot|s) - \pi_\mathsf{T}(\cdot|s) \right\|_1^2 \right] }
$$

$$
\overset{(ii)}{\leq} \sqrt{ \overline{C}_\rho^2 \cdot \mathbb{E}_\mathsf{T} \mathbb{E}_{s \sim \rho_{\mathsf{T}-1}^*} \left[ 2 D_{\mathrm{KL}} \left( \pi_\mathsf{T}^*(\cdot|s) \| \pi_\mathsf{T}(\cdot|s) \right) \right] }
$$

$$
= \sqrt{ \overline{C}_\rho^2 \cdot 2 \mathbb{E}_\mathsf{T} \left[ \sigma_\pi^\mathsf{T} \right] }
$$

$$
\overset{(iii)}{\lesssim} \frac{1}{\sqrt{\lambda}} \left( \frac{\sqrt{\log T}}{T^{1/5}} + \sqrt{\varepsilon} \right),
$$

where step $(i)$ follows from Cauchy-Schwarz inequality, step $(ii)$ follows from Assumption 4 and Pinsker's inequality, and step $(iii)$ follows from the bound in equation (27). The above equation, together with Jensen's inequality, proves equation (13). We have completed the proof of Theorem 1.

### B.1  PROOF OF LEMMA 6

The following lemma characterizes this policy improvement step. The proof is provided in Section D.4.

**Lemma 8.** *For any distributions $p^*, p \in \Delta(\mathcal{A})$, state $s \in \mathcal{S}$ and function $G : \mathcal{S} \times \mathcal{A} \to \mathbb{R}$, it holds for $p' \in \Delta(\mathcal{A})$ with $p'(\cdot) \propto p(\cdot) \cdot \exp\left[ \alpha G(s, \cdot) \right]$ that*

$$
D_{\mathrm{KL}}\left( p^* \| p' \right) \leq D_{\mathrm{KL}}\left( p^* \| p \right) - \alpha \left\langle G(s, \cdot), p^* - p \right\rangle + \alpha^2 \left\| G(s, \cdot) \right\|_\infty^2 / 2.
$$

Recall that

$$
\widehat{\pi}_{t+1}(\cdot|s) \propto \pi_t(\cdot|s) \cdot \exp\left[ \alpha_t \left( \widehat{Q}_t^\lambda(s, \cdot) - \lambda \log \pi_t(\cdot|s) \right) \right].
$$

Lemma 8 implies that for each $s \in \mathcal{S}$, we have

$$D_{\mathrm{KL}}\left(\pi_{t+1}^*(\cdot|s)\|\widehat{\pi}_{t+1}(\cdot|s)\right)$$

$$\leq D_{\mathrm{KL}}\left(\pi_{t+1}^*(\cdot|s)\|\pi_t(\cdot|s)\right) - \alpha_t \left\langle \widehat{Q}_t^\lambda(s,\cdot) - \lambda \log \pi_t(\cdot|s), \pi_{t+1}^*(\cdot|s) - \pi_t(\cdot|s) \right\rangle + \left\|\widehat{Q}_t^\lambda\right\|_\infty^2 \alpha_t^2/2$$

$$= D_{\mathrm{KL}}\left(\pi_{t+1}^*(\cdot|s)\|\pi_t(\cdot|s)\right) - \alpha_t \left\langle Q_t^\lambda(s,\cdot) - \lambda \log \pi_t(\cdot|s), \pi_{t+1}^*(\cdot|s) - \pi_t(\cdot|s) \right\rangle$$

$$+ \alpha_t \left\langle Q_t^\lambda(s,\cdot) - \widehat{Q}_t^\lambda(s,\cdot), \pi_{t+1}^*(\cdot|s) - \pi_t(\cdot|s) \right\rangle + \left\|\widehat{Q}_t^\lambda\right\|_\infty^2 \alpha_t^2/2$$

$$\leq D_{\mathrm{KL}}\left(\pi_{t+1}^*(\cdot|s)\|\pi_t(\cdot|s)\right) - \alpha_t \left\langle Q_t^\lambda(s,\cdot) - \lambda \log \pi_t(\cdot|s), \pi_{t+1}^*(\cdot|s) - \pi_t(\cdot|s) \right\rangle$$

$$+ 2\alpha_t \left\|Q_t^\lambda(s,\cdot) - \widehat{Q}_t^\lambda(s,\cdot)\right\|_\infty + \left\|\widehat{Q}_t^\lambda\right\|_\infty^2 \alpha_t^2/2.$$

Recall that $\pi_{t+1}(\cdot|s) = (1-\eta)\widehat{\pi}_{t+1}(\cdot|s) + \frac{\eta}{|\mathcal{A}|}\mathbf{1}_{|\mathcal{A}|}$. Lemma 2 implies that

$$D_{\mathrm{KL}}\left(\pi_{t+1}^*(\cdot|s)\|\pi_{t+1}(\cdot|s)\right)$$

$$\leq D_{\mathrm{KL}}\left(\pi_{t+1}^*(\cdot|s)\|\widehat{\pi}_{t+1}(\cdot|s)\right) + 2\eta. \tag{29}$$

$$\leq D_{\mathrm{KL}}\left(\pi_{t+1}^*(\cdot|s)\|\pi_t(\cdot|s)\right) - \alpha_t \left\langle Q_t^\lambda(s,\cdot) - \lambda \log \pi_t(\cdot|s), \pi_{t+1}^*(\cdot|s) - \pi_t(\cdot|s) \right\rangle$$

$$+ \underbrace{2\alpha_t \left\|Q_t^\lambda(s,\cdot) - \widehat{Q}_t^\lambda(s,\cdot)\right\|_\infty + \left\|\widehat{Q}_t^\lambda\right\|_\infty^2 \alpha_t^2/2 + 2\eta}_{Y_t(s)}. \tag{30}$$

Taking expectation over $\rho_t^*$ on both sides of (30) yields

$$\mathbb{E}_{\rho_t^*}\left[D_{\mathrm{KL}}\left(\pi_{t+1}^*\|\pi_{t+1}\right)\right]$$

$$\leq \mathbb{E}_{\rho_t^*}\left[D_{\mathrm{KL}}\left(\pi_{t+1}^*\|\pi_t\right)\right] - \alpha_t \mathbb{E}_{s\sim\rho_t^*}\left[\left\langle Q_t^\lambda(s,\cdot) - \lambda \log \pi_t(\cdot|s), \pi_{t+1}^*(\cdot|s) - \pi_t(\cdot|s)\right\rangle\right] + \mathbb{E}_{s\sim\rho_t^*}\left[Y_t(s)\right]$$

$$\overset{(a)}{=} \mathbb{E}_{\rho_t^*}\left[D_{\mathrm{KL}}\left(\pi_{t+1}^*\|\pi_t\right)\right] - (1-\gamma)\alpha_t \left[J_{\mu_t}^\lambda(\pi_{t+1}^*) - J_{\mu_t}^\lambda(\pi_t)\right] - \alpha_t \lambda \mathbb{E}_{\rho_t^*}\left[D_{\mathrm{KL}}\left(\pi_{t+1}^*\|\pi_t\right)\right] + \mathbb{E}_{s\sim\rho_t^*}\left[Y_t(s)\right]$$

$$\overset{(b)}{\leq} (1-\alpha_t\lambda)\mathbb{E}_{\rho_t^*}\left[D_{\mathrm{KL}}\left(\pi_{t+1}^*\|\pi_t\right)\right] + \mathbb{E}_{s\sim\rho_t^*}\left[Y_t(s)\right]$$

$$\overset{(c)}{\leq} (1-\alpha_t\lambda)\underbrace{\mathbb{E}_{\rho_t^*}\left[D_{\mathrm{KL}}\left(\pi_t^*\|\pi_t\right)\right]}_{B_1} + (1-\alpha_t\lambda)\underbrace{\left|\mathbb{E}_{\rho_t^*}\left[D_{\mathrm{KL}}\left(\pi_{t+1}^*\|\pi_t\right) - D_{\mathrm{KL}}\left(\pi_t^*\|\pi_t\right)\right]\right|}_{B_2} + \mathbb{E}_{s\sim\rho_t^*}\left[Y_t(s)\right],$$

$$\tag{31}$$

where step (a) follows from Lemma 4; step (b) follows from the fact that $J_{\mu_t}^\lambda(\pi_t) \leq J_{\mu_t}^\lambda(\pi_{t+1}^*)$, as $\pi_{t+1}^* = \Gamma_1^\lambda(\mu_t)$ is the optimal policy for the regularized $\mathrm{MDP}_{\mu_t}$; and step (c) holds due to triangle inequality.

Next we bound the first and second terms on the RHS of (31) separately.

- For the second term $B_2$: Note that $\pi_{t+1}^*$ and $\pi_t^*$ are the optimal policy for the regularized $\mathrm{MDP}_{\mu_t}$ and $\mathrm{MDP}_{\mu_{t-1}}$, respectively. Define

$$\tau := \frac{1}{|\mathcal{A}|}\exp\left(-\frac{R_{\max} + \gamma\lambda\log|\mathcal{A}|}{\lambda(1-\gamma)}\right).$$

  By Lemma 5, for all $(s,a) \in \mathcal{S} \times \mathcal{A}$, we have

$$\pi_{t+1}^*(a|s) \geq \tau, \text{ and } \pi_t^*(a|s) \geq \tau.$$

  Applying Lemma 3 yields

$$B_2 \leq \kappa\mathbb{E}_{s\sim\rho_t^*}\left[\left\|\pi_t^*(\cdot|s) - \pi_{t+1}^*(\cdot|s)\right\|_1\right]$$

$$= \kappa\mathbb{E}_{s\sim\rho^*}\left[\frac{\rho_t^*(s)}{\rho^*(s)} \cdot \left\|\pi_t^*(\cdot|s) - \pi_{t+1}^*(\cdot|s)\right\|_1\right]$$

$$\leq \kappa C_\rho\mathbb{E}_{s\sim\rho^*}\left[\left\|\pi_t^*(\cdot|s) - \pi_{t+1}^*(\cdot|s)\right\|_1\right] \qquad \text{Assumption 4}$$

$$= \kappa C_\rho D\left(\Gamma_1^\lambda(\mu_{t-1}), \Gamma_1^\lambda(\mu_t)\right)$$

$$\leq \kappa C_\rho d_1 \left\|\mu_{t-1} - \mu_t\right\|_{\mathcal{H}}, \qquad \text{Assumption (2)} \qquad (32)$$

where

$$\kappa := 1 + \log \frac{1}{\min\left\{\tau, \frac{\eta}{|\mathcal{A}|}\right\}}$$

$$\leq 2 \max\left\{\log \frac{|\mathcal{A}|}{\eta}, \frac{2}{1-\gamma}\log|\mathcal{A}| + \frac{R_{\max}}{\lambda(1-\gamma)}\right\}$$

$$\leq \frac{4}{1-\gamma}\log\frac{|\mathcal{A}|}{\eta} + \frac{2R_{\max}}{\lambda(1-\gamma)}$$

$$= \frac{4}{1-\gamma}\mathrm{KL}_{\max} + \frac{2R_{\max}}{\lambda(1-\gamma)}.$$

- For the first term $B_1$: We have

$$B_1 = \mathbb{E}_{\rho^*_{t-1}}\left[D_{\mathrm{KL}}\left(\pi^*_t\|\pi_t\right)\right] + \left(\mathbb{E}_{\rho^*_t} - \mathbb{E}_{\rho^*_{t-1}}\right)\left[D_{\mathrm{KL}}\left(\pi^*_t\|\pi_t\right)\right]$$

$$= \mathbb{E}_{\rho^*_{t-1}}\left[D_{\mathrm{KL}}\left(\pi^*_t\|\pi_t\right)\right] + \mathbb{E}_{s\sim\rho^*}\left[\frac{\rho^*_t(s) - \rho^*_{t-1}(s)}{\rho^*(s)}D_{\mathrm{KL}}\left(\pi^*_t(\cdot|s)\|\pi_t(\cdot|s)\right)\right]$$

$$\overset{(a)}{\leq} \mathbb{E}_{\rho^*_{t-1}}\left[D_{\mathrm{KL}}\left(\pi^*_t\|\pi_t\right)\right] + \mathbb{E}_{s\sim\rho^*}\left[\frac{|\rho^*_t(s) - \rho^*_{t-1}(s)|}{\rho^*(s)}\right] \cdot \mathrm{KL}_{\max},$$

$$\overset{(b)}{\leq} \mathbb{E}_{\rho^*_{t-1}}\left[D_{\mathrm{KL}}\left(\pi^*_t\|\pi_t\right)\right] + \mathrm{KL}_{\max} \cdot d_0 \left\|\mu_t - \mu_{t-1}\right\|_{\mathcal{H}} \tag{33}$$

where step (a) uses the fact that $D_{\mathrm{KL}}\left(\pi^*_t(\cdot|s)\|\pi_t(\cdot|s)\right) \leq \mathrm{KL}_{\max} := \log\frac{|\mathcal{A}|}{\eta}$ (cf. Lemma 2) and step (b) follows from Assumption 5.

Combining (31), (32) and (33), we have

$$\mathbb{E}_{\rho^*_t}\left[D_{\mathrm{KL}}\left(\pi^*_{t+1}\|\pi_{t+1}\right)\right]$$

$$\leq (1 - \lambda\alpha_t)\mathbb{E}_{\rho^*_{t-1}}\left[D_{\mathrm{KL}}\left(\pi^*_t\|\pi_t\right)\right]$$

$$+ (1 - \lambda\alpha_t)d_0 \cdot \mathrm{KL}_{\max}\left\|\mu_t - \mu_{t-1}\right\|_{\mathcal{H}} + (1 - \lambda\alpha_t)\kappa C_\rho d_1 \left\|\mu_{t-1} - \mu_t\right\|_{\mathcal{H}} + \mathbb{E}_{s\sim\rho^*_t}\left[Y_t(s)\right]$$

$$= (1 - \lambda\alpha_t)\mathbb{E}_{\rho^*_{t-1}}\left[D_{\mathrm{KL}}\left(\pi^*_t\|\pi_t\right)\right]$$

$$+ (1 - \lambda\alpha_t)\left(d_0 \cdot \mathrm{KL}_{\max} + \kappa C_\rho d_1\right)\left\|\mu_{t-1} - \mu_t\right\|_{\mathcal{H}} + \mathbb{E}_{s\sim\rho^*_t}\left[Y_t(s)\right]. \tag{34}$$

Note that

$$\mathbb{E}_{s\sim\rho^*_t}\left[Y_t(s)\right] = 2\alpha_t\mathbb{E}_{s\sim\rho^*_t}\left[\left\|Q^\lambda_t(s,\cdot) - \widehat{Q}^\lambda_t(s,\cdot)\right\|_\infty\right] + \frac{\left\|\widehat{Q}^\lambda_t\right\|^2_\infty}{2}\alpha^2_t + 2\eta$$

$$\leq 2\alpha_t\sqrt{\mathbb{E}_{s\sim\rho^*_t}\left[\left\|Q^\lambda_t(s,\cdot) - \widehat{Q}^\lambda_t(s,\cdot)\right\|^2_\infty\right]} + \frac{\left\|\widehat{Q}^\lambda_t\right\|^2_\infty}{2}\alpha^2_t + 2\eta$$

$$\leq 2\varepsilon\alpha_t + \frac{Q^2_{\max}}{2}\alpha^2_t + 2\eta,$$

where the last step holds by the assumption on the policy evaluation error and the fact that $\widehat{Q}^\lambda_{t-1} : \mathcal{S} \times \mathcal{A} \to [0, Q_{\max}]$ satisfies $\left\|\widehat{Q}^\lambda_{t-1}\right\|_\infty \leq Q_{\max}$ by definition. Combining the last two display equations proves the lemma.

### B.2 PROOF OF LEMMA 7

*Proof.* According to the update rule (10) for the embedded mean-field state, we have

$$
\begin{aligned}
& \left\|\mu_{t+1} - \mu^*\right\|_{\mathcal{H}} \\
&= \left\|(1 - \beta_t)\mu_t + \beta_t\Gamma_2(\pi_{t+1}, \mu_t) - \mu^*\right\|_{\mathcal{H}} \\
&= \left\|(1 - \beta_t)\left(\mu_t - \mu^*\right) + \beta_t\left(\Gamma_2\left(\Gamma_1^\lambda(\mu_t), \mu_t\right) - \mu^*\right) - \beta_t\left[\Gamma_2\left(\Gamma_1^\lambda(\mu_t), \mu_t\right) - \Gamma_2(\pi_{t+1}, \mu_t)\right]\right\|_{\mathcal{H}} \\
&\leq (1 - \beta_t)\left\|(\mu_t - \mu^*)\right\|_{\mathcal{H}} + \beta_t\left\|\Gamma_2\left(\Gamma_1^\lambda(\mu_t), \mu_t\right) - \mu^*\right\|_{\mathcal{H}} \\
&\qquad + \beta_t\left\|\Gamma_2\left(\Gamma_1^\lambda(\mu_t), \mu_t\right) - \Gamma_2(\pi_{t+1}, \mu_t)\right\|_{\mathcal{H}} \\
&\overset{(i)}{=} (1 - \beta_t)\left\|\mu_t - \mu^*\right\|_{\mathcal{H}} + \beta_t\underbrace{\left\|\Gamma_2\left(\Gamma_1^\lambda(\mu_t), \mu_t\right) - \Gamma_2\left(\Gamma_1^\lambda(\mu^*), \mu^*\right)\right\|_{\mathcal{H}}}_{(a)} \\
&\qquad + \beta_t\underbrace{\left\|\Gamma_2\left(\Gamma_1^\lambda(\mu_t), \mu_t\right) - \Gamma_2(\pi_{t+1}, \mu_t)\right\|_{\mathcal{H}}}_{(b)},
\end{aligned}
\tag{35}
$$

where the equality $(i)$ follows from the fact that $\mu^* = \Gamma_2\left(\Gamma_1^\lambda(\mu^*), \mu^*\right)$.

Lemma 1 implies that $\Lambda(\mu) = \Gamma_2\left(\Gamma_1^\lambda(\mu), \mu\right)$ is $d_1 d_2 + d_3$ Lipschitz. It follows that

$$
(a) \leq (d_1 d_2 + d_3)\left\|\mu_t - \mu^*\right\|_{\mathcal{H}}.
\tag{36}
$$

By Assumption 3, we have

$$
(b) \leq d_2 D\left(\Gamma_1^\lambda(\mu_t), \pi_{t+1}\right).
\tag{37}
$$

Combining Eqs. (35)-(37) yields

$$
\left\|\mu_{t+1} - \mu^*\right\|_{\mathcal{H}} \leq \left(1 - \beta_t\overline{d}\right)\left\|\mu_t - \mu^*\right\|_{\mathcal{H}} + d_2\beta_t D\left(\Gamma_1^\lambda(\mu_t), \pi_{t+1}\right),
\tag{38}
$$

where $\overline{d} = 1 - d_1 d_2 - d_3 > 0$.

Let us bound the second RHS term above. By the definition of policy distance $D$ in equation (12), we have

$$
\begin{aligned}
D\left(\Gamma_1^\lambda(\mu_t), \pi_{t+1}\right) &= \mathbb{E}_{\rho^*}\left[\left\|\Gamma_1^\lambda(\mu_t) - \pi_{t+1}\right\|_1\right] \\
&= \mathbb{E}_{s \sim \rho^*}\left[\left\|\pi_{t+1}^*(\cdot|s) - \pi_{t+1}(\cdot|s)\right\|_1\right] \\
&= \mathbb{E}_{s \sim \rho_t^*}\left[\frac{\rho^*(s)}{\rho_t^*(s)}\left\|\pi_{t+1}^*(\cdot|s) - \pi_{t+1}(\cdot|s)\right\|_1\right] \\
&\leq \left\{\mathbb{E}_{s \sim \rho_t^*}\left[\left|\frac{\rho^*(s)}{\rho_t^*(s)}\right|^2\right] \cdot \mathbb{E}_{s \sim \rho_t^*}\left[\left\|\pi_{t+1}^*(\cdot|s) - \pi_{t+1}(\cdot|s)\right\|_1^2\right]\right\}^{1/2} \\
&\leq \overline{C}_\rho\sqrt{\mathbb{E}_{s \sim \rho_t^*}\left[D_{\mathrm{KL}}\left(\pi_{t+1}^*(\cdot|s)\|\pi_{t+1}(\cdot|s)\right)\right]},
\end{aligned}
\tag{39}
$$

where the first inequality holds due to Cauchy-Schwartz inequality, the last inequality follows from Assumption 4 and Pinsker's inequality.

Combining (38)-(39) gives

$$
\left\|\mu_{t+1} - \mu^*\right\|_{\mathcal{H}} \leq \left(1 - \beta_t\overline{d}\right)\left\|\mu_t - \mu^*\right\|_{\mathcal{H}} + d_2\beta_t\overline{C}_\rho\sqrt{\mathbb{E}_{s \sim \rho_t^*}\left[D_{\mathrm{KL}}\left(\pi_{t+1}^*(\cdot|s)\|\pi_{t+1}(\cdot|s)\right)\right]}.
$$

This completes the proof. $\qquad\square$

## C    PROOF OF COROLLARY 1

*Proof.* Note that for each $t \in [T]$, we have

$$
\begin{aligned}
D(\pi_t, \pi^*) &\leq D\left(\pi_t, \pi_t^*\right) + D\left(\pi_t^*, \pi^*\right) \\
&= D\left(\pi_t, \pi_t^*\right) + D\left(\Gamma_1^\lambda(\mu_t), \Gamma_1^\lambda(\mu^*)\right) \\
&\leq D\left(\pi_t, \pi_t^*\right) + d_1\left\|\mu_t - \mu^*\right\|_{\mathcal{H}},
\end{aligned}
$$

where the last step follows from Assumption 2 on the Lipschitzness of $\Gamma_1^\lambda$. It follows that

$$D\left(\frac{1}{T}\sum_{t=1}^{T}\pi_t, \pi^*\right) + \left\|\frac{1}{T}\sum_{t=1}^{T}\mu_t - \mu^*\right\|_{\mathcal{H}}$$

$$\leq \frac{1}{T}\sum_{t=1}^{T}D\left(\pi_t, \pi^*\right) + \frac{1}{T}\sum_{t=1}^{T}\|\mu_t - \mu^*\|_{\mathcal{H}}$$

$$\leq \frac{1}{T}\sum_{t=1}^{T}\left(D\left(\pi_t, \pi_t^*\right) + d_1\|\mu_t - \mu^*\|_{\mathcal{H}}\right) + \frac{1}{T}\sum_{t=1}^{T}\|\mu_t - \mu^*\|_{\mathcal{H}}$$

$$\lesssim \frac{1}{\sqrt{\lambda}}\left(\frac{\sqrt{\log T}}{T^{1/5}} + \sqrt{\varepsilon}\right),$$

where in the last step we apply the bounds (13) and (14) in Theorem 1. □

## D  ADDITIONAL PROOFS

### D.1  PROOF OF LEMMA 1

*Proof.* By the definition of $\Lambda$, we have

$$\left\|\Lambda^\lambda(\mu) - \Lambda^\lambda(\mu')\right\|_{\mathcal{H}}$$

$$= \left\|\Gamma_2\left(\Gamma_1^\lambda(\mu), \mu\right) - \Gamma_2\left(\Gamma_1^\lambda(\mu'), \mu'\right)\right\|_{\mathcal{H}}$$

$$\leq \left\|\Gamma_2\left(\Gamma_1^\lambda(\mu), \mu\right) - \Gamma_2\left(\Gamma_1^\lambda(\mu'), \mu\right)\right\|_{\mathcal{H}} + \left\|\Gamma_2\left(\Gamma_1^\lambda(\mu'), \mu\right) - \Gamma_2\left(\Gamma_1^\lambda(\mu'), \mu'\right)\right\|_{\mathcal{H}} \qquad \text{triangle inequality}$$

$$\leq d_2 D\left(\Gamma_1^\lambda(\mu), \Gamma_1^\lambda(\mu')\right) + d_3\|\mu - \mu'\|_{\mathcal{H}} \qquad\qquad\qquad\qquad\qquad \text{Assumption 3}$$

$$\leq d_1 d_2\|\mu - \mu'\|_{\mathcal{H}} + d_3\|\mu - \mu'\|_{\mathcal{H}}, \qquad\qquad\qquad\qquad\qquad\qquad \text{Assumption 2}$$

which proves the lemma. □

### D.2  PROOF OF LEMMA 4

*Proof.* By the definition of $V_\mu^{\lambda,\pi}$ in (4), we have

$$V_\mu^{\lambda,\pi'}(s)$$

$$= \mathbb{E}_{a_t \sim \pi'(s_t), s_{t+1} \sim \mathrm{P}(\cdot|s_t, a_t, \mu)}\left[\sum_{t=0}^{\infty}\gamma^t\left[r_\mu^{\lambda,\pi'}(s, a) + V_\mu^{\lambda,\pi}(s_t) - V_\mu^{\lambda,\pi}(s_t)\right] \mid s_0 = s\right].$$

$$= \mathbb{E}_{a_t \sim \pi'(s_t), s_{t+1} \sim \mathrm{P}(\cdot|s_t, a_t, \mu)}\left[\sum_{t=0}^{\infty}\gamma^t\left[r_\mu^{\lambda,\pi'}(s, a) + \gamma V_\mu^{\lambda,\pi}(s_{t+1}) - V_\mu^{\lambda,\pi}(s_t)\right] \mid s_0 = s\right] + V_\mu^{\lambda,\pi}(s).$$

$$(40)$$

Recall that the Q-function $Q_\mu^{\lambda,\pi}$ of a policy $\pi$ for the regularized MDP$_\mu$ is related to $V_\mu^{\lambda,\pi}$ as

$$V_\mu^{\lambda,\pi}(s) = \mathbb{E}_{a\sim\pi(s)}\left[Q_\mu^{\lambda,\pi}(s, a) - \lambda\log\pi(a|s)\right] = \left\langle Q_\mu^{\lambda,\pi}(s, \cdot), \pi(\cdot|s)\right\rangle + \lambda\mathbb{H}\left(\pi(\cdot|s)\right), \qquad \forall s \in \mathcal{S},$$

$$Q_\mu^{\lambda,\pi}(s, a) = r(s, a, \mu) + \gamma\mathbb{E}_{s_1\sim\mathrm{P}(\cdot|s, a, \mu)}\left[V_\mu^{\lambda,\pi}(s_1)\right], \qquad \forall(s, a) \in \mathcal{S} \times \mathcal{A}.$$

We have

$$\left\langle Q_\mu^{\lambda,\pi}(s, \cdot), \pi'(\cdot|s)\right\rangle = \mathbb{E}_{a\sim\pi'(s)}\left[Q_\mu^{\lambda,\pi}(s, a)\right],$$

$$= \mathbb{E}_{a\sim\pi'(s)}\left[r(s, a, \mu) + \gamma\mathbb{E}_{s_1\sim\mathrm{P}(\cdot|s, a, \mu)}\left[V_\mu^{\lambda,\pi}(s_1)\right]\right]$$

$$= \mathbb{E}_{a\sim\pi'(s), s_1\sim\mathrm{P}(\cdot|s, a, \mu)}\left[r_\mu^{\lambda,\pi'}(s, a) + \gamma V_\mu^{\lambda,\pi}(s_1) + \lambda\log\pi'(a|s)\right]$$

$$= \mathbb{E}_{a\sim\pi'(s), s_1\sim\mathrm{P}(\cdot|s, a, \mu)}\left[r_\mu^{\lambda,\pi'}(s, a) + \gamma V_\mu^{\lambda,\pi}(s_1)\right] - \lambda\mathbb{H}\left(\pi'(\cdot|s)\right).$$

Therefore,

$$\left\langle Q_\mu^{\lambda,\pi}(s,\cdot), \pi'(\cdot|s) - \pi(\cdot|s)\right\rangle$$

$$=\mathbb{E}_{a\sim\pi'(s),s_1\sim P(\cdot|s,a,\mu)}\left[r^{\lambda,\pi'}(s,a,\mu) + \gamma V_\mu^{\lambda,\pi}(s_1)\right] - \lambda\mathbb{H}\left(\pi'(\cdot|s)\right) - V_\mu^{\lambda,\pi}(s) + \lambda\mathbb{H}\left(\pi(\cdot|s)\right)$$

$$=\mathbb{E}_{a\sim\pi'(s),s_1\sim P(\cdot|s,a,\mu)}\left[r^{\lambda,\pi'}(s,a,\mu) + \gamma V_\mu^{\lambda,\pi}(s_1) - V_\mu^{\lambda,\pi}(s)\right] - \lambda\left[\mathbb{H}\left(\pi'(\cdot|s)\right) - \mathbb{H}\left(\pi(\cdot|s)\right)\right].$$

$$(41)$$

Plugging (41) into (40), we have

$$V_\mu^{\lambda,\pi'}(s) - V_\mu^{\lambda,\pi'}(s)$$

$$=\mathbb{E}_{a_t\sim\pi'(s_t),s_{t+1}\sim P(\cdot|s_t,a_t,\mu)}\left[\sum_{t=0}^\infty \gamma^t \left\langle Q_\mu^{\lambda,\pi}(s_t,\cdot), \pi'(\cdot|s_t) - \pi(\cdot|s_t)\right\rangle \mid s_0 = s\right]$$

$$+ \mathbb{E}_{a_t\sim\pi'(s_t),s_{t+1}\sim P(\cdot|s_t,a_t,\mu)}\left[\sum_{t=0}^\infty \gamma^t \lambda\left(\mathbb{H}\left(\pi'(\cdot|s_t)\right) - \mathbb{H}\left(\pi(\cdot|s_t)\right)\right) \mid s_0 = s\right]. \quad (42)$$

Recall the definition of $J_\mu^\lambda(\pi)$ in (5). Taking expectation with respect to $s \sim \nu_0$ on both sides of (42) yields $\qquad\square$

$$J_\mu^\lambda(\pi') - J_\mu^\lambda(\pi)$$

$$=\mathbb{E}_{s_0\sim\nu_0,a_t\sim\pi'(s_t),s_{t+1}\sim P(\cdot|s_t,a_t,\mu)}\left[\sum_{t=0}^\infty \gamma^t \left\langle Q_\mu^{\lambda,\pi}(s_t,\cdot), \pi'(\cdot|s_t) - \pi(\cdot|s_t)\right\rangle\right]$$

$$+ \mathbb{E}_{s_0\sim\nu_0,a_t\sim\pi'(s_t),s_{t+1}\sim P(\cdot|s_t,a_t,\mu)}\left[\sum_{t=0}^\infty \gamma^t \lambda\left(\mathbb{H}\left(\pi'(\cdot|s_t)\right) - \mathbb{H}\left(\pi(\cdot|s_t)\right)\right)\right]$$

$$=\frac{1}{1-\gamma}\mathbb{E}_{s\sim\rho_\mu^{\pi'}}\left[\left\langle Q_\mu^{\lambda,\pi}(s,\cdot), \pi'(\cdot|s) - \pi(\cdot|s)\right\rangle + \lambda\left(\mathbb{H}\left(\pi'(\cdot|s)\right) - \mathbb{H}\left(\pi(\cdot|s)\right)\right)\right]. \quad (43)$$

For the entropy term in (43), we have

$$\mathbb{E}_{s\sim\rho_\mu^{\pi'}}\left[\mathbb{H}\left(\pi'(\cdot|s)\right) - \mathbb{H}\left(\pi(\cdot|s)\right)\right]$$

$$=\mathbb{E}_{s\sim\rho_\mu^{\pi'}}\left[\left\langle\log\frac{1}{\pi'(\cdot|s)}, \pi'(\cdot|s)\right\rangle - \left\langle\log\frac{1}{\pi(\cdot|s)}, \pi(\cdot|s)\right\rangle\right]$$

$$=\mathbb{E}_{s\sim\rho_\mu^{\pi'}}\left[\left\langle\log\frac{1}{\pi(\cdot|s)} - \log\frac{\pi'(\cdot|s)}{\pi(\cdot|s)}, \pi'(\cdot|s)\right\rangle - \left\langle\log\frac{1}{\pi(\cdot|s)}, \pi(\cdot|s)\right\rangle\right]$$

$$=\mathbb{E}_{s\sim\rho_\mu^{\pi'}}\left[\left\langle\log\frac{1}{\pi(\cdot|s)}, \pi'(\cdot|s) - \pi(\cdot|s)\right\rangle - D_{\mathrm{KL}}\left(\pi'(\cdot|s)\|\pi(\cdot|s)\right)\right]. \quad (44)$$

Taking (44) into (43) yields the desired equation in Lemma 4.

### D.3 PROOF OF LEMMA 5

*Proof.* Note that the value function $V_\mu^{\lambda,\pi}$ can be written as

$$V_\mu^{\lambda,\pi}(s) = \mathbb{E}\left[\sum_{t=0}^\infty \gamma^t r_\mu^{\lambda,\pi}(s_t,a_t)|s_0 = s\right].$$

By the definition of $r_\mu^{\lambda,\pi}$ in (1), we have $0 \le \mathbb{E}_\pi\left[r_\mu^{\lambda,\pi}(s_t,a_t)\right] \le R_{\max} + \lambda\log|\mathcal{A}|$. Therefore,

$$0 \le V_\mu^{\lambda,\pi}(s) \le \frac{R_{\max} + \lambda\log|\mathcal{A}|}{1-\gamma}, \qquad \forall s \in \mathcal{S},$$

and

$$0 \le Q_\mu^{\lambda,\pi}(s,a) \le R_{\max} + \gamma\frac{R_{\max} + \lambda\log|\mathcal{A}|}{1-\gamma} = \frac{R_{\max} + \gamma\lambda\log|\mathcal{A}|}{1-\gamma}, \qquad \forall s \in \mathcal{S}, a \in \mathcal{A}.$$

For the second inequality, we have

$$
\begin{aligned}
\pi_\mu^{\lambda,*}(a|s) &= \frac{\exp\left(Q_\mu^{\lambda,*}(s,a)/\lambda\right)}{\sum_{b\in\mathcal{A}} \exp\left(Q_\mu^{\lambda,*}(s,b)/\lambda\right)} \\
&\geq \frac{1}{\sum_{b\in\mathcal{A}} \exp\left(Q_{\max}/\lambda\right)} = \frac{1}{e^{Q_{\max}/\lambda}|\mathcal{A}|}
\end{aligned}
$$

as claimed. $\qquad\square$

### D.4 PROOF OF LEMMA 8

*Proof.* For any function $g : \mathcal{A} \to \mathbb{R}$ and distribution $p \in \Delta(\mathcal{A})$, let $z : \mathcal{A} \to \mathbb{R}$ be a constant function defined by

$$
z(a) = \log\left(\sum_{a'\in\mathcal{A}} p(a') \cdot \exp\left(\alpha g(a')\right)\right).
$$

Note that for any distributions $p^*, p' \in \Delta(\mathcal{A}), \langle z, p^* - p'\rangle = 0$. Since

$$
p'(\cdot) \propto p(\cdot) \cdot \exp\left(\alpha g(\cdot)\right),
$$

we have $\alpha g(\cdot) = z(\cdot) + \log(p'(\cdot)/p(\cdot))$. Hence

$$
\begin{aligned}
\alpha\langle g, p^* - p'\rangle &= \langle z + \log(p'/p), p^* - p'\rangle \\
&= \langle z, p^* - p'\rangle + \langle\log(p^*/p), p^*\rangle + \langle\log(p'/p^*), p^*\rangle + \langle\log(p'/p), -p'\rangle \\
&= D_{\mathrm{KL}}\left(p^*\|p\right) - D_{\mathrm{KL}}\left(p^*\|p'\right) - D_{\mathrm{KL}}\left(p'\|p\right).
\end{aligned}
$$

Therefore, for each state $s \in \mathcal{S}$, we have

$$
\begin{aligned}
\alpha\langle G(s,\cdot), p^* - p\rangle &= \alpha\langle G(s,\cdot), p^* - p'\rangle + \alpha\langle G(s,\cdot), p' - p\rangle \\
&= D_{\mathrm{KL}}\left(p^*\|p\right) - D_{\mathrm{KL}}\left(p^*\|p'\right) - D_{\mathrm{KL}}\left(p'\|p\right) + \alpha\langle G(s,\cdot), p' - p\rangle \\
&\leq D_{\mathrm{KL}}\left(p^*\|p\right) - D_{\mathrm{KL}}\left(p^*\|p'\right) - D_{\mathrm{KL}}\left(p'\|p\right) + \alpha\|G(s,\cdot)\|_\infty \cdot \|p - p'\|_1.
\end{aligned}
$$

Rearranging terms yields

$$
D_{\mathrm{KL}}\left(p^*\|p'\right) \leq D_{\mathrm{KL}}\left(p^*\|p\right) - \alpha\langle G(s,\cdot), p^* - p\rangle - D_{\mathrm{KL}}\left(p'\|p\right) + \alpha\|G(s,\cdot)\|_\infty \cdot \|p - p'\|_1. \quad (45)
$$

Meanwhile, by Pinsker's inequality, it holds that

$$
D_{\mathrm{KL}}\left(p'\|p\right) \geq \|p - p'\|_1^2/2. \quad (46)
$$

By combining (45) and (46), we obtain

$$
\begin{aligned}
D_{\mathrm{KL}}\left(p^*\|p'\right) &\leq D_{\mathrm{KL}}\left(p^*\|p\right) - \alpha\langle G(s,\cdot), p^* - p\rangle - \|p - p'\|_1^2/2 + \alpha\|G(s,\cdot)\|_\infty \cdot \|p - p'\|_1 \\
&\leq D_{\mathrm{KL}}\left(p^*\|p\right) - \alpha\langle G(s,\cdot), p^* - p\rangle + \alpha^2\|G(s,\cdot)\|_\infty^2/2,
\end{aligned}
$$

which concludes the proof. $\qquad\square$

## E   A WEAKER ASSUMPTION ON CONCENTRABILITY

In this section, we consider a weaker assumption on concentrability, under which Algorithm 1 learns a policy-population pair that is $\widetilde{O}(T^{-1/9})$-approximate NE after $T$ iterations.

We consider the following distance metric between two policies $\pi, \pi' \in \Pi$:

$$
W(\pi, \pi') := \sqrt{\mathbb{E}_{s\sim\rho^*}\left[\|\pi(\cdot|s) - \pi'(\cdot|s)\|_1^2\right]}. \quad (47)
$$

Similarly as before, we assume certain Lipschitz properties for the two mappings $\Gamma_1^\lambda : \mathcal{M} \to \Pi$ and $\Gamma_2 : \Pi \times \mathcal{M} \to \mathcal{M}$ defined in Section 2.3. In particular, we impose the following two assumtpions, both stated in terms of the new distance metric $W(\cdot, \cdot)$ defined in (47) above.

**Assumption 6.** *There exists a constant $d_1 > 0$, such that for any $\mu, \mu' \in \mathcal{M}$, it holds that*

$$
W\left(\Gamma_1^\lambda(\mu), \Gamma_1^\lambda(\mu')\right) \leq d_1\|\mu - \mu'\|_{\mathcal{H}}.
$$

**Assumption 7.** *There exist constants $d_2 > 0, d_3 > 0$ such that for any policies $\pi, \pi' \in \Pi$ and embedded mean-field states $\mu, \mu' \in \mathcal{M}$, it holds that*

$$\|\Gamma_2(\pi, \mu) - \Gamma_2(\pi', \mu)\|_{\mathcal{H}} \leq d_2 W(\pi, \pi'),$$
$$\|\Gamma_2(\pi, \mu) - \Gamma_2(\pi, \mu')\|_{\mathcal{H}} \leq d_3 \|\mu - \mu'\|_{\mathcal{H}}.$$

Assumptions 6 and 7 immediately imply Lipschitzness of the composite mapping $\Lambda^\lambda : \mathcal{M} \to \mathcal{M}$, which we recall is defined as $\Lambda^\lambda(\mu) = \Gamma_2\left(\Gamma_1^\lambda(\mu), \mu\right)$.

**Lemma 9.** *Suppose Assumptions 6 and 7 hold. Then for each $\mu, \mu' \in \mathcal{M}$, it holds that*

$$\left\|\Lambda^\lambda(\mu) - \Lambda^\lambda(\mu')\right\|_{\mathcal{H}} \leq (d_1 d_2 + d_3) \|\mu - \mu'\|_{\mathcal{H}}.$$

We also consider the following relaxed, $\ell_2$-type assumption on the concentrability coefficients.

**Assumption 8** (Finite Concentrability Coefficients)**.** *There exist two constants $C_\rho, \overline{C}_\rho > 0$ such that for each $\mu \in \mathcal{M}$, it holds that*

$$\left\{ \mathbb{E}_{s \sim \rho_\mu^{\pi_\mu^{\lambda,*}}} \left[ \left| \frac{\rho_\mu^{\pi_\mu^{\lambda,*}}(s)}{\rho^*(s)} \right|^2 \right] \right\}^{1/2} \leq C_\rho \qquad and \qquad \left\{ \mathbb{E}_{s \sim \rho_\mu^{\pi_\mu^{\lambda,*}}} \left[ \left| \frac{\rho^*(s)}{\rho_\mu^{\pi_\mu^{\lambda,*}}(s)} \right|^2 \right] \right\}^{1/2} \leq \overline{C}_\rho.$$

We establish the following convergence result for Algorithm 1.

**Theorem 2.** *Suppose that Assumptions 1, 5, 6, 7, and 8 hold and $d_1 d_2 + d_3 < 1$ and that the error in the policy evaluation step in Algorithm 1 satisfies*

$$\mathbb{E}_{s \sim \rho_t^*} \left[ \left\| Q_t^\lambda(s, \cdot) - \widehat{Q}_t^\lambda(s, \cdot) \right\|_\infty^2 \right] \leq \varepsilon^2, \qquad \forall t \in [T].$$

*With the choice of*

$$\eta = c_\eta T^{-1}, \qquad \alpha_t \equiv \alpha = c_\alpha T^{-4/9}, \qquad \beta_t \equiv \beta = c_\beta T^{-8/9},$$

*for some universal constants $c_\eta > 0$, $c_\alpha > 0$ and $c_\beta > 0$ in Algorithm 1, the resulting policy and embedded mean-field state sequence $\{(\pi_t, \mu_t)\}_{t=1}^T$ satisfy*

$$W\left(\frac{1}{T}\sum_{t=1}^T \pi_t, \frac{1}{T}\sum_{t=1}^T \pi_t^*\right) \leq \frac{1}{T}\sum_{t=1}^T W(\pi_t, \pi_t^*) \lesssim \frac{1}{\lambda^{1/4}}\left(\frac{(\log T)^{1/4}}{T^{1/9}} + \varepsilon^{1/4}\right), \qquad (48)$$

$$\left\|\frac{1}{T}\sum_{t=1}^T \mu_t - \mu^*\right\|_{\mathcal{H}} \leq \frac{1}{T}\sum_{t=1}^T \|\mu_t - \mu^*\|_{\mathcal{H}} \lesssim \frac{1}{\lambda^{1/4}}\left(\frac{(\log T)^{1/4}}{T^{1/9}} + \varepsilon^{1/4}\right). \qquad (49)$$

The following corollary of Theorem 2 shows that after $T$ iterations of our algorithm, the average policy-population pair $\left(\frac{1}{T}\sum_{t=1}^T \pi_t, \frac{1}{T}\sum_{t=1}^T \mu_t\right)$ is an $\widetilde{\mathcal{O}}\left(T^{-1/9}\right)$-approximate NE.

**Corollary 2.** *Under the assumptions of Theorem 2, we have*

$$W\left(\frac{1}{T}\sum_{t=1}^T \pi_t, \pi^*\right) + \left\|\frac{1}{T}\sum_{t=1}^T \mu_t - \mu^*\right\|_{\mathcal{H}} \lesssim \frac{1}{\lambda^{1/4}}\left(\frac{(\log T)^{1/4}}{T^{1/9}} + \varepsilon^{1/4}\right).$$

### E.1 PROOFS OF THEOREM 2 AND COROLLARY 2

The proof follows similar lines as those of Theorem 1 and Corollary 1, with all appearances of the distance $D$ replaced by the new distance $W$. Below we only point out the modifications needed.

Lemma 6 remains valid as stated. For the proof of this lemma, the only different step is bounding the term $B_2$ in equation (31). In particular, the bounds in equation (32) should be replaced by the

following:

$$
\begin{aligned}
B_2 &\leq \kappa \mathbb{E}_{s \sim \rho_t^*} \left[ \left\| \pi_t^*(\cdot|s) - \pi_{t+1}^*(\cdot|s) \right\|_1 \right] \\
&= \kappa \mathbb{E}_{s \sim \rho^*} \left[ \frac{\rho_t^*(s)}{\rho^*(s)} \cdot \left\| \pi_t^*(\cdot|s) - \pi_{t+1}^*(\cdot|s) \right\|_1 \right] \\
&\leq \kappa \sqrt{ \mathbb{E}_{s \sim \rho^*} \left[ \left( \frac{\rho_t^*(s)}{\rho^*(s)} \right)^2 \right] \cdot \mathbb{E}_{s \sim \rho^*} \left[ \left\| \pi_t^*(\cdot|s) - \pi_{t+1}^*(\cdot|s) \right\|_1^2 \right] } \\
&\leq \kappa C_\rho \cdot \sqrt{ \mathbb{E}_{s \sim \rho^*} \left[ \left\| \pi_t^*(\cdot|s) - \pi_{t+1}^*(\cdot|s) \right\|_1^2 \right] } \qquad &\text{Assumption 8} \\
&= \kappa C_\rho W \left( \Gamma_1^\lambda(\mu_{t-1}), \Gamma_1^\lambda(\mu_t) \right) \\
&\leq \kappa C_\rho d_1 \left\| \mu_{t-1} - \mu_t \right\|_{\mathcal{H}}. \qquad &\text{Assumption 6} \qquad (50)
\end{aligned}
$$

Lemma 7 should be replaced by the following lemma.

**Lemma 10.** *Under the setting of Theorem 2, for each $t \geq 0$, we have*

$$
\sigma_\mu^{t+1} \leq \left( 1 - \beta_t \overline{d} \right) \sigma_\mu^t + d_2 \sqrt{\overline{C}_\rho} \beta_t \left( \sigma_\pi^{t+1} \right)^{1/4},
$$

*where $\overline{d} = 1 - d_1 d_2 - d_3 > 0$.*

The proof of Lemma 10 is similar to that of Lemma 7. The only different step is the term $D \left( \Gamma_1^\lambda(\mu_t), \pi_{t+1} \right)$ in equation (38) should be replaced by $W \left( \Gamma_1^\lambda(\mu_t), \pi_{t+1} \right)$, which can be bounded as follows:

$$
\begin{aligned}
W \left( \Gamma_1^\lambda(\mu_t), \pi_{t+1} \right) &= \sqrt{ \mathbb{E}_{s \sim \rho^*} \left[ \left\| \pi_{t+1}^*(\cdot|s) - \pi_{t+1}(\cdot|s) \right\|_1^2 \right] } \\
&= \sqrt{ \mathbb{E}_{s \sim \rho_t^*} \left[ \frac{\rho^*(s)}{\rho_t^*(s)} \left\| \pi_{t+1}^*(\cdot|s) - \pi_{t+1}(\cdot|s) \right\|_1^2 \right] } \\
&\leq \left\{ \mathbb{E}_{s \sim \rho_t^*} \left[ \left| \frac{\rho^*(s)}{\rho_t^*(s)} \right|^2 \right] \cdot \mathbb{E}_{s \sim \rho_t^*} \left[ \left\| \pi_{t+1}^*(\cdot|s) - \pi_{t+1}(\cdot|s) \right\|_1^4 \right] \right\}^{1/4} \\
&\overset{(i)}{\lesssim} \sqrt{\overline{C}_\rho} \cdot \left\{ \mathbb{E}_{s \sim \rho_t^*} \left[ \left\| \pi_{t+1}^*(\cdot|s) - \pi_{t+1}(\cdot|s) \right\|_1^2 \right] \right\}^{1/4} \\
&\overset{(ii)}{\lesssim} \sqrt{\overline{C}_\rho} \left\{ \mathbb{E}_{s \sim \rho_t^*} \left[ D_{\mathrm{KL}} \left( \pi_{t+1}^*(\cdot|s) \| \pi_{t+1}(\cdot|s) \right) \right] \right\}^{1/4}. \qquad (51)
\end{aligned}
$$

where step $(i)$ holds by Assumption 8 and the fact that $\| \nu - \nu' \|_1 \leq 2, \forall \nu, \nu' \in \Delta(\mathcal{A})$, and step $(ii)$ follows Pinsker's inequality.

We now turn to the proof of Theorem 2.

We first establish the convergence for $\sigma_\pi^t$ by following the exactly same steps from equation (21) up to equation (25). We restate the bound on $\frac{1}{T} \sum_{t=0}^{T-1} \sigma_\pi^t$ in (25) as follows:

$$
\frac{1}{T} \sum_{t=0}^{T-1} \sigma_\pi^t \leq \frac{1}{T \lambda \alpha} \sigma_\pi^0 + \frac{\overline{C}_1 \beta}{\lambda \alpha} + \frac{2\varepsilon}{\lambda} + \frac{Q_{\max}^2}{2\lambda} \alpha + \frac{2\eta}{\lambda \alpha}. \qquad (52)
$$

When choosing $\alpha = \mathcal{O}(T^{-4/9})$, $\beta = \mathcal{O}(T^{-8/9})$ and $\eta = \mathcal{O}(T^{-1})$, we have $\overline{C}_1 = \mathcal{O}(\log T)$. Therefore, we obtain

$$
\frac{1}{T} \sum_{t=0}^{T-1} \sigma_\pi^t \lesssim \frac{\log T}{\lambda T^{4/9}} + \frac{2\varepsilon}{\lambda}. \qquad (53)
$$

If we let $\mathsf{T}$ be a random number sampled uniformly from $\{1, \ldots, T\}$, then the above equation can be written equivalently as

$$
\mathbb{E}_{\mathsf{T}} \left[ \sigma_\pi^{\mathsf{T}} \right] \lesssim \frac{\log T}{\lambda T^{4/9}} + \frac{2\varepsilon}{\lambda}. \qquad (54)
$$

We now proceed to bound the average embedded mean-field state $\frac{1}{T}\sum_{t=0}^{T-1}\sigma_\mu^t$. Lemma 10 implies

$$\sigma_\mu^t \le \frac{1}{\overline{d}\beta_t}\left(\sigma_\mu^t - \sigma_\mu^{t+1}\right) + \frac{d_2\sqrt{\overline{C}_\rho}}{\overline{d}}\left(\sigma_\pi^{t+1}\right)^{1/4}. \tag{55}$$

With $\beta_t \equiv \beta = \mathcal{O}(T^{-8/9})$, averaging equation (55) over iteration $t = 0, \ldots, T-1$, we obtain

$$\frac{1}{T}\sum_{t=0}^{T-1}\sigma_\mu^t \le \frac{1}{\overline{d}\beta T}\left(\sigma_\mu^0 - \sigma_\mu^T\right) + \frac{d_2\sqrt{\overline{C}_\rho}}{\overline{d}T}\sum_{t=0}^{T-1}\left(\sigma_\pi^{t+1}\right)^{1/4}$$

$$\le \frac{\sigma_\mu^0}{\overline{d}\beta T} + \frac{d_2\sqrt{\overline{C}_\rho}}{\overline{d}T}\sum_{t=0}^{T-1}\left(\sigma_\pi^{t+1}\right)^{1/4}$$

$$\overset{(i)}{\le} \frac{\sigma_\mu^0}{\overline{d}\beta T} + \frac{d_2\sqrt{\overline{C}_\rho}}{\overline{d}}\sqrt{\frac{1}{T}\sum_{t=0}^{T-1}\sqrt{\sigma_\pi^{t+1}}}$$

$$\overset{(ii)}{\le} \frac{\sigma_\mu^0}{\overline{d}\beta T} + \frac{d_2\sqrt{\overline{C}_\rho}}{\overline{d}}\left(\frac{1}{T}\sum_{t=0}^{T-1}\sigma_\pi^{t+1}\right)^{1/4}$$

where steps $(i)$ and $(ii)$ follow from Cauchy-Schwarz inequality.

From equation (53), we have

$$\frac{1}{T}\sum_{t=0}^{T-1}\sigma_\mu^t \lesssim \frac{\sigma_\mu^0}{\overline{d}}T^{-1/9} + \frac{d_2\sqrt{\overline{C}_\rho}}{\overline{d}}\left(\frac{\log T}{\lambda T^{4/9}} + \frac{2\varepsilon}{\lambda}\right)^{1/4}$$

$$\lesssim \left(\frac{\log T}{\lambda T^{4/9}} + \frac{2\varepsilon}{\lambda}\right)^{1/4}$$

$$\lesssim \frac{1}{\lambda^{1/4}}\left(\frac{(\log T)^{1/4}}{T^{1/9}} + \varepsilon^{1/4}\right).$$

This equation, together with Jensen's inequality, proves equation (49) in Theorem 2.

Turning to equation (48) in Theorem 2, we have

$$\frac{1}{T}\sum_{t=1}^{T}W\left(\pi_t, \pi_t^*\right) = \mathbb{E}_\mathsf{T}\left[W\left(\pi_\mathsf{T}, \pi_\mathsf{T}^*\right)\right]$$

$$= \mathbb{E}_\mathsf{T}\sqrt{\mathbb{E}_{s\sim\rho^*}\left[\|\pi_\mathsf{T}^*(\cdot|s) - \pi_\mathsf{T}(\cdot|s)\|_1^2\right]}$$

$$\overset{(i)}{\le} \sqrt{\mathbb{E}_\mathsf{T}\mathbb{E}_{s\sim\rho_{\mathsf{T}-1}^*}\left[\frac{\rho^*(s)}{\rho_{\mathsf{T}-1}^*(s)}\|\pi_\mathsf{T}^*(\cdot|s) - \pi_\mathsf{T}(\cdot|s)\|_1^2\right]}$$

$$\overset{(ii)}{\le} \left\{\mathbb{E}_\mathsf{T}\mathbb{E}_{s\sim\rho_{\mathsf{T}-1}^*}\left[\left|\frac{\rho^*(s)}{\rho_{\mathsf{T}-1}^*(s)}\right|^2\right] \cdot \mathbb{E}_\mathsf{T}\mathbb{E}_{s\sim\rho_{\mathsf{T}-1}^*}\left[\|\pi_\mathsf{T}^*(\cdot|s) - \pi_\mathsf{T}(\cdot|s)\|_1^4\right]\right\}^{1/4}$$

$$\overset{(iii)}{\lesssim} \left\{\overline{C}_\rho^2 \cdot \mathbb{E}_\mathsf{T}\mathbb{E}_{s\sim\rho_{\mathsf{T}-1}^*}\left[\|\pi_\mathsf{T}^*(\cdot|s) - \pi_\mathsf{T}(\cdot|s)\|_1^2\right]\right\}^{1/4}$$

$$\overset{(iv)}{\lesssim} \sqrt{\overline{C}_\rho} \cdot \left\{\mathbb{E}_\mathsf{T}\mathbb{E}_{s\sim\rho_{\mathsf{T}-1}^*}\left[D_{\mathrm{KL}}\left(\pi_\mathsf{T}^*(\cdot|s)\|\pi_\mathsf{T}(\cdot|s)\right)\right]\right\}^{1/4}$$

$$= \sqrt{\overline{C}_\rho} \cdot \left\{\mathbb{E}_\mathsf{T}\left[\sigma_\pi^\mathsf{T}\right]\right\}^{1/4}$$

$$\overset{(v)}{\lesssim} \frac{1}{\lambda^{1/4}}\left(\frac{(\log T)^{1/4}}{T^{1/9}} + \varepsilon^{1/4}\right),$$

where step $(i)$ holds due to Jensen's inequality, step $(ii)$ follows from Cauchy-Schwarz inequality, step $(iii)$ follows from Assumption 8 and the fact that $\|\nu - \nu'\|_1 \le 2, \forall\nu, \nu' \in \Delta(\mathcal{A})$, step $(iv)$

comes from Pinsker's inequality, and step $(v)$ follows from the bound in equation (54). The above equation, together with Jensen's inequality, proves equation (48). We have completed the proof of Theorem 2.

The proof of Corollary 2 is the same as that of Corollary 1 and is omitted here.

