# OpenReview forum: "Provable Fictitious Play for General Mean-Field Games"
_ICLR.cc/2021/Conference — Reject_

### Official Review · AnonReviewer3 · 2020-10-23
**Overall solid work but with unclear novelty, that needs to be better motivated**

**Rating:** 5
**Confidence:** 3

**Review:**

The paper considers a Markov mean-field game where the continuum of players is summarized by their distribution over the state space. The policy of each player is a function of the state alone (and not a function of mean-field distribution). It is assumed that the game is "well-behaved" so it can be embedded in a regular and bounded kernel.  It is also assumed the relevant that the composition of the relevant mappings forms a contraction mapping, along with several other technical assumptions. An algorithm that approximates the Nash equilibrium (NE) of the game is proposed, which consists of an entropy-regulated reinforcement learning step and an update of the  (embedded) mean-field state step. Based on the contraction structure, it is proved that this algorithm converges to an approximate NE with a rate (T^-1/5) for T iterations, and the approximation depends on the entropy regularization parameter and the error bound of the reinforcement learning step.

The paper is clearly written and the math is solid and well explained. The topic of the paper is likely to be interesting to a broad audience, provided that more insight and discussion are given. The main concern is with the novelty or significance of the result.

Single Loop/Double Loop -

i) While I agree that a single loop is overall more elegant, this feels like some kind of an intermediate motivation for the paper. The real motivation is of course the computational complexity of the NE approximation algorithm. It is not obvious that a single loop algorithm is less complicated overall since it can take more outer iteration to converge. Providing complexity guarantees and comparing them with state of the art, or at least providing some numerical simulations for the same purpose, seems necessary to make the contribution significant. Since you do provide the convergence rate, converting that to complexity seems straightforward, isn't it?

ii) It is claimed that (Guo et al., 2019) and others require to solve an MDP each iteration. The abstract also implies this ("in stark contrast..."). Is this really the case? it looks like (Guo et al., 2019) also works with an approximate solution to the MDP, it's just that the number of RL iterations it has to do each time has a lower bound. Please discuss that in more detail, since not being clear regarding this issue makes the whole single loop advantage vague. For example, if an algorithm runs only 20 iterations of RL every outer iteration, is it really a double loop algorithm? and what if it runs 2 iterations?  if it ends up being a quantitative comparison, then one would naturally wonder how good of an epsilon (in Theorem 1) can be achieved with a "one iteration" approximation of Q (aren't iterations required to approximate Q?).  If it's big, and \lambda is small, then the approximation of the NE might be poor.

Kernel -

The writing is a bit confusing regarding whatever this is a technique or an assumption, and I think it's more of the latter. To clarify this, please add to Assumption 1 the part that says that the game can be embedded in this kernel. Nevertheless, I do agree it is a reasonable assumption to make.

Entropy Regulation -

It is not entirely clear why this regulation is crucial, and the price is convergence to an approximate NE instead of the exact one (which the abstract is misleading about). What happens if there are multiple NE? (is this even the case?) It seems reasonable to request to converge to one of them. How much faster does it make the convergence, compared to other algorithms? For example, it looks like (Guo et al., 2019) don't employ this regularization and prove that the NE is unique, so why is it needed here? why can't we choose \lambda=0? More discussion is needed to make this design choice to look less arbitrary.

Reinforcement Learning and Fictitious Play  -

I find the use of both names a bit misleading, at least in some contexts.

RL - While the algorithm definitely calls an RL routine, the algorithm itself isn't RL in the sense that it isn't model free. It looks like the step in (10) requires knowing the transition probabilities to compute (7). This seems like a disappointing assumption. How does that compare to the literature? can this assumption be removed? Intuitively it feels like making this step "noisy" as well should be possible as long as the contraction structure remains. Please correct me if didn't get this part right.

Fictitious Play (FP) plays the *best* response to the empirical distribution of the past actions of the opponents. This isn't exactly the case here, and the title+abstract make it look like the contribution is to study the behavior of FP in a mean-field game, and not to solve a Mean-field game with an algorithm that has some resemblance to FP.

Contraction - stating the assumption as d1d2+d3<1 is a bit too dry. Isn't that possible just to assume that the composition in Lemma 1 is a contraction? it reads better.

Assumptions - I think that providing more discussion and intuition about each of the assumptions will make the paper more interesting. What assumptions will be extremely hard to remove,  and what is more for convenience? please also mention the literature in this discussion with more detail than "x was assumed in y", for example, what was required in other papers that made other assumptions. A conclusion section is missing in this paper, and this could be the right place to suggest future directions that can make other researchers interested in picking up on your result and perhaps remove some of the assumptions.

Organization - the order of the lemmas in the appendix interrupts the flow of the reading. It's best to introduce a lemma and then prove it, giving some intuition about its role in the bigger picture. Alternatively, some lemmas (statement + proof) can be postponed, and the proofs that use them can say "where (a) follows from Lemma x".

Minor Comments:

"the proposed algorithm converges to the Nash equilibrium of the MFG"  - but the NE of the MFG (with no regularization) isn't necessarily unique. Please rephrase.

"\mathcal{A} is finite" this is more than a technical assumption, it means that the paper considers discrete action sets, which seems like something that should be stated more clearly.

I think it's better to reverse (7), since the second equation is the mapping.

Some parameters, like \lambda, are missing from the algorithm's input, and also from the Theorem's statement.

Why does the proof of Lemma 2 need Holder's inequality at the end? looks like it's just bounded by 2\eta?

I believe that a factor of |A| is missing in \kappa of Lemma 6.

(32) - should be Assumption 2 and not Assumption (2).

below (34) - why Q of t-1 and not t?

---

> ### Author Response · Authors · 2020-11-25
> **Response to Reviewer 3**
>
> Thank you very much for your detailed feedback and constructive comments. Please see our responses below.
>
> Q1: Single loop vs double loop
> A1: Our single loop algorithm only requires policy evaluation for the current policy, while a double loop algorithm requires finding the **optimal** policy of the MDP. Finding the optimal policy is difficult (if not infeasible) unless the transition kernel $ P$ of the MDP is known or there is a simulator for sampling from $P( \cdot |s,a)$ for arbitrary $a$. Such samples are generally not available since the policy and the mean-field, and hence the induced MDP, change in every step of the game. On the other hand, policy evaluation for the current policy $\pi_t$ only requires sampling from $P( \cdot | s, \pi_t(s) )$. As we discussed in response to Reviewer 1 Q1, such samples are available from a **single** step of the mean-field game, as we observe the transitions of a large number of agents.
>
> Q2: Kernel: Is it a technique or an assumption that the game can be embedded in the kernel?
>
> A2: It is an assumption that the game can be embedded in an RKHS space. We will add this in the revision. This assumption is similar in spirit to recent work on RL with function approximation, where it is assumed that the value function, transition kernel and/or policy of the MDP (approximately) lie in some function class (say linear functions).
>
> Q3: It is not entirely clear why entropy regularization is crucial.
>
> A3: Entropy regularization has been used extensively in MDPs. Recent work has shown that such regularization can accelerate the convergence of policy gradient algorithms (Cen et al., 2020; Shani et al., 2019). In particular, policy gradient algorithms can converge linearly when computing optimal value functions of the regularized MDP --- a significant improvement over the non-regularized setting. Of course, entropy regularization induces extra approximation error. We can choose an appropriate regularization parameter $\lambda$ to balance the error and convergence rate. Please also see our response to Q2 of Reviewer 1.
>
> Q4: Reinforcement Learning and Fictitious Play---I find the use of both names a bit misleading, at least in some contexts.
>
> A4: We consider the RL setting where the transition kernel is unknown, in which case step (10) in our algorithm can be approximately computed using samples; see our response to Q1.
>
> Indeed, we are using the term Fictitious Play in a more liberal way: our algorithm computes the soft-min of the Q function, whereas the best response corresponds to the hard/exact minimum. Therefore, instead of playing the exact best response to the empirical actions of the opponent, we play a good response while ensuring the policy does not change too quickly. Nevertheless, we believe our algorithm captures the key ingredient of FP, namely, trying to infer/predict the behavior of the opponent based on its past actions.
>
>
> Q5: Contraction and assumption
>
> A5: These are good suggestions. We will revise accordingly.
>
> Q6: Organization and minor comments
>
> A6: Thank you for other detailed comments. We will address these comments accordingly in the revision.

---

### Official Review · AnonReviewer2 · 2020-10-26
**Official Blind Review**

**Rating:** 5
**Confidence:** 3

**Review:**

Summary:
This paper proposes a single loop reinforcement learning algorithm for stationary mean-field games. In particular, when viewing the mean-field state and the stationary policy as two players, the goal is to find a Nash equilibrium. Updating these two players alternatively via gradient-descent and proximal policy optimization, the proposed algorithm is proved to converge sublinearly to the Nash equilibrium.

Comment:
This paper is well-written as it well explains the problem setting of the mean-field game step-by-step. While existing fixed-point algorithms require to solve the MDP induced by the current mean-field state exactly, the authors propose to take the model-free RL approach which is able to handle continuous state space. Also, the single loop structure of the algorithm is often favored over the nested loop scheme in terms of the practical performance (which unfortunately is not validated through empirical study in this paper).

Concern:
My major concern is the assumption made on the relations between three Lipschitz continuity constants in Theorem 1: To achieve the described convergence rate, the authors assume that $d_1 d_2 + d_3<1$. I do not see how such configuration would hold in general. Such an assumption significantly limits the applicability of the theory developed in the paper (which is main claim of the paper as there is no empirical study provided).

---

> ### Author Response · Authors · 2020-11-25
> **Response to Reviewer 2**
>
> Thank you very much for your detailed feedback and constructive comments. Please see our responses below.
>
> Q1: Assumption on d1d2 + d3 < 1
>
> A1: Indeed some recent work investigates sufficient conditions on the system parameters (transition kernel and reward function) of the mean-field game for the Lipschitzness of the induced operators $\Gamma_1(\cdot)$ and $\Gamma_2(\cdot)$. For the regularized mean-field game considered in our paper, recent work by Anatarci et al. [1] gives sufficient conditions on the Lipschitzness property of composition operator $\Gamma(\cdot).$ The assumption $d_1d_2+d_3<1$ holds if the transition kernel and reward function are Lipschitz and the corresponding Lipschitz constants satisfy a certain condition.
>
> [1] Berkay Anahtarci, Can Deha Kariksiz, Naci Saldi, “Q-Learning in Regularized Mean-field Games”, arxiv, https://arxiv.org/abs/2003.12151 2020.
>
>
>
> Q2: Lack of empirical study
>
> A2: This is an excellent suggestion. Thank you. We will include a numerical result in the revision.

---

### Official Review · AnonReviewer4 · 2020-11-01
**Review on "Provable Fictitious Play for General Mean-Field Games"**

**Rating:** 3
**Confidence:** 5

**Review:**



Summary:
The paper focuses on the computation of Nash equilibrium in a Multi agent setting, with a very large number of agents. This leads to the consideration of reinforcement leaning algorithms for mean field games (i.e. with an infinite number of agents) and obtain the convergence of a single loop fictititious play algorithm to the Nash equilibrium of entropy regularized mean field games.

Strength of the paper:
- The connection between the derived mathematical property and the very interesting problem of interest, i.e. learning in a multi agent setting with numerous agents is well motivated.
- The paper is well written and technically sound. The mathematical results provided are technical, difficult and are presented in a pedagogic manner

Weakness of the paper:
- The convergence property relies on a very strong and a priori unrealistic assumption on the Lipschitz coefficients os the distribution <-> policy mappings: d_1d_2+d_3<1. Authors should justify when this assumption can indeed be satisfied and directly read on underlying Mean Field Game Structure.
- No numerical experiments are presented to verify the convergence of the algorithm and compare it to alternative methods in the literature.

Recommandation; Given the scope of the paper in comparison to the one of the ICLR conference, together with the weaknesses detailed above, I recommend to reject the paper.

Questions during the rebuttal period:
1. Can you provide reasonable sufficient conditions on the mean field game dynamics ensuring that the assumption d_1d_2+d_3<1 is satisfied? Even though it already appeared in previous papers, it is really very strong stated as such.
2. Can you provide numerical experiments confirming the theoretical findings of the paper?
3. Focusing on penalized MFGs, the role of the penalization coefficient \lambda seems to be of particular interest, and a necessity in your approach. Is the dependence of the NE (policy and or state distribution) clear with respect to \lambda? What are the practical implications of the choice of \lambda?
4. How does your approach compare to the 2-time scale approach provided by Angiuli et al. (Unified Reinforcement Q-Learning for Mean Field Game and ControlProblems) in terms of single/multiple loops?

---

> ### Author Response · Authors · 2020-11-25
> **Response to Reviewer 4**
>
> Response to Reviewer 4
>
> Thank you very much for your detailed feedback and constructive comments. Please see our responses below.
>
> Q1: Can you provide reasonable sufficient conditions on the mean-field game dynamics ensuring that the assumption d_1d_2+d_3<1 is satisfied?
>
> A1: Indeed some recent work investigates sufficient conditions on the system parameters (transition kernel and reward function) of the mean-field game for the Lipschitzness of the induced operators $\Gamma_1(\cdot)$ and $\Gamma_2(\cdot)$. For the regularized mean-field game considered in our paper, recent work by Anatarci et al. [1] gives sufficient conditions on the Lipschitzness property of composition operator $\Gamma(\cdot).$ The assumption $d_1d_2+d_3<1$ holds if the transition kernel and reward function are Lipschitz and the corresponding Lipschitz constants satisfy a certain condition.
>
> [1] Berkay Anahtarci, Can Deha Kariksiz, Naci Saldi, “Q-Learning in Regularized Mean-field Games”, arxiv, https://arxiv.org/abs/2003.12151 2020.
>
>
> Q2: Provide numerical experiments confirming the theoretical findings of the paper.
>
> A3: This is an excellent suggestion. Thank you. We will add a numerical simulation in the revision.
>
> Q3: Focusing on penalized MFGs, the role of the penalization coefficient \lambda seems to be of particular interest, and a necessity in your approach. Is the dependence of the NE (policy and or state distribution) clear with respect to \lambda? What are the practical implications of the choice of \lambda?
>
> A3: This is a very good question. Indeed we can character the regularization error. In particular, the approximation error of the regularized NE for the original unregularized NE scales proportionally with $\lambda$ (cf. the last equation on page 5). On the other hand, the convergence rate to the regularized NE of our algorithm scales inverse proportionally
> with $\sqrt(\lambda)$, implying that convergence can be accelerated with a higher level of entropy regularization. Therefore, it is desirable to choose the regularization parameter $\lambda$ that balances the target accuracy level and convergence rate. See also A2 in the response to Reviewer 1.
>
> Q4: How does your approach compare to the 2-time scale approach provided by Angiuli et al. (Unified Reinforcement Q-Learning for Mean Field Game and ControlProblems) in terms of single/multiple loops?
>
> A4: Thank you very much for pointing this reference. Indeed the two-time scale approach by [Angiuli et al.] is also a single-loop algorithm. Their approach updates Q-function in each iteration like Q-learning and can be applied to learn mean-field games and control problems. We would like to remark that we provide a rigorous analysis of the convergence rate of our approach, while the guarantee of their approach is not clear.

---

### Official Review · AnonReviewer1 · 2020-11-02
**Contribution to an important problem but with some non-negligible limitations**

**Rating:** 5
**Confidence:** 5

**Review:**

This paper considers reinforcement learning (RL) for finding Nash equilibriums (NE) in general stationary mean-field games (MFG). It attempts to address the problem of designing a single-loop RL algorithm which updates the policy and the mean-field state simultaneously in each iteration while maintaining provable global convergence to NEs, which is a significant open problem in the existing literature. To this end, the authors propose a PPO-based fictitious play algorithm and show that it converges to approximate NEs at a sublinear rate.

The major contributions of this paper are listed below:
1. Unlike the existing works in the literature, which solves the mean-field state induced RL subproblem to near optima in each iteration before moving on to the next mean-field update, the algorithm proposed in this paper only performs a single step of policy improvement and then immediately moves on to the mean-field update step.
2. By introducing entropy regularization, the authors also makes the Lipschitz continuity of the optimal policies with respect to the mean-field states (Assumption 2) more reasonable (due to the softmax format and uniqueness of the optimal policies) than the counterparts of this assumption for unregularized MDPs in the literature.
3. it introduces kernel mean embedding, which adds to the flexibility of the proposed framework.

However, this paper also has several limitations and weaknesses, as listed below:
1. The most significant issue is that in the propose algorithm (Algorithm 1), the step in line 3 requires computing an approximate evaluation of $\hat{Q}_t^{\lambda}$. Although the authors explain that this could be done using TD(0), LSTD and so on, all these algorithms (and in general, any RL algorithm) still require fixing the mean-field state $\mathcal{L}_t$ while simulating through and getting samples from the MDP subproblem induced by $\mathcal{L}_t$. In practice, the mean-field state cannot be easily fixed if one is interacting with the real-world (instead of a simulator), as when the policy is executed the mean-field gets updated immediately as well. This is the actual source of the limitations of the existing double-loop algorithms, which still cannot be avoided by the proposed algorithm in this paper. An actual single-loop algorithm should not require fixing the mean-field state. In particular, for Algorithm 1, only one update should be allowed when evaluating the $\hat{Q}_t^{\lambda}$ function. So the actual contribution in terms of proposing a single-loop algorithm is limited.
2. The paper does not include the errors caused by introducing the entropy regularization term to the original problem into the final results. Although it should not be very difficult to include these errors in the final results due to the bound at the bottom of page 5 and the Lipschitz continuity assumptions made throughout the paper, the authors should formally include it and discuss about the trade-off between choosing a larger $\lambda$ to improve the convergence rate and choosing a smaller $\lambda$ to control the regularization error.
3. The necessity of introduction of kernel mean embedding and the corresponding assumptions on RKHS is questionable. By embedding the original mean-field state $\mathcal{L}_t$ as $\mu_t$, we are still dealing with an infinite dimensional mean-field state embedding ($\mu_t$), which does not seem to introduce any quantitative simplification (apart from the qualitative “lower complexity” claim mentioned by the authors). From the presentation of this paper, it seems that the major use of the embedding is introducing a well-defined distance inherited from the underlying RKHS. However, such tools can also be achieved by considering Wasserstein distances and TV distances for general probability measures. The authors should provide a clear explanation on why introducing the mean embedding is necessary and helpful.
4. On a related point, in line 5 of Algorithm 1, unless $\mathcal{M}$ is shown to be convex, the update (10) may no longer yield $\mu_{t+1}$ in $\mathcal{M}$, and in such a case the corresponding original mean-field state $\mathcal{L}_{t+1}$ cannot be retrieved and hence the algorithm cannot move on (cf. (7)). The authors should explain clearly why $\mathcal{M}$ is convex or why this is not an issue.
5. In general, when the state space is infinite, the Q-table will also be infinite. In such a case, either function approximation or something like the nearest neighbor Q-learning [2] should be used. The authors should at least discuss about this for line 3 in Algorithm 1.

Finally, some slightly more minor comments:
1. The concentrability coefficients look different from the literature, which typically has the denominator being the initial distribution or a uniform bound on discounted state-visitation measures corresponding to arbitrary policies (see e.g., [1, Assumption 1]) instead of the optimal one (like the $\rho^\star$ here). The authors may want to provide some further intuition and explanations on why this is the case to help the readers gain better insights.
2. The performance difference lemma (Lemma 4) seems to be mostly the same as Lemma 1 (Performance improvement) in Cen et al., 2020. The authors should either directly refer to Cen et al., 2020 for this result, or state clearly the difference.
3. The authors assumed that the kernel of the RKHS satisfies that $k(s,s)\leq 1$ for all $s\in\mathcal{S}$. How stringent is this assumption? Is it always achievable by some rescaling? If so, the authors may want to add a quick comment on this in the draft.
4. In formula (3), it seems that there is a missing $\lambda$ coefficient before the regularization term $\mathbb{H}$.
5. In the abstract, “by the iterates mean-field states” might better be “by the iterates of mean-field states”.

In short, this paper attempts to address an important problem in the RL for MFG literature, and made some successful contributions. However, due to the technical limitations and weaknesses mentioned above, I think the paper is still not ready for publication in ICLR in its current format.

[1] Shani, Lior, Yonathan Efroni, and Shie Mannor. "Adaptive trust region policy optimization: Global convergence and faster rates for regularized mdps." arXiv preprint arXiv:1909.02769 (2019).

[2] Shah, Devavrat, and Qiaomin Xie. "Q-learning with nearest neighbors." In Advances in Neural Information Processing Systems, pp. 3111-3121. 2018.

---

> ### Author Response · Authors · 2020-11-25
> **Response to Reviewer 1**
>
> Thank you very much for your detailed feedback and constructive comments. Kindly see our detailed responses below.
>
> Q1: A potential issue of computing an approximate Q-function in the algorithm is that it requires fixing the mean-field state.
>
> A1: This is an excellent question. Although it is tempting to view solving mean-field game (MFG) as solving a sequence of MDPs, each indexed by the mean-field state $\mathcal{L_t}$, this view is a bit limited as it only focuses on the representative agent and neglects the fact that each agent of the MFG has observations. Just consider the case with $N$ agents and we let the local state and action of the $i$-th agent at the $t$-th timestep, $\forall i \in [N]$, be denoted by $s_t^t$ and $a_t^i$. Then, the mean-field state, $L_t$, is the empirical measure of   $ \{ s_t^i \}_{ i \in  [N] }$. Enforcing permutation invariance, we equip each agent with the same policy $\pi_t$.Then, at the $t$-th timestep, the $i$-th agent takes an action $a_t^i \in \pi_t (\cdot \,|\, s_t)$, observes a reward $r_t^i = r( s_t^i, a_t^i, \mathcal{L}_t)$  and next state  $ s _{t+1} ^t \sim   P(\cdot \, | \, s_t^i, a_t^i, \mathcal{L}_t)$. Then, we have $N$ transition data just for the $t$-th timestep, i.e., $\{ (s_t^i, a_t^i, r_t^i, s _{t+1}^i), i \in [N]\}$. The nice thing is that these $N$ transition tuple follows from the same policy $\pi_t$ and the same transition kernel $P(\cdot \, | \, \cdot, \cdot \mathcal{L}_t)$. Therefore, we can use these $N$ transition tuples to run a standard policy evaluation algorithm.
>
>
> Therefore, when utilizing the feedback information of all the agents of the MFG, policy evaluation step in our algorithm actually does not require simulating samples from the MDP induced by the fixed mean-field state --- a **single** transition of the mean-field game acts as a “simulator”.
>
> Furthermore, such an interesting observation also holds for the infinite-agent case. In this case, we just need to sample from the mean-field state. Specifically, in the $t$-th timestep, the mean-field state is $\mathcal{L}_t$ such that the state of the representative agent $s_t$, has distribution $\mathcal{L}_t$. So generate data from the *single* transition of mean-field game, for any $N > 0$, we can draw $N$ i.i.d. observations $s_t^i \sim \mathcal{L}_t$, $\forall i \in [N]$, sample action $a_t^i \sim \pi_t (\cdot \, | \, s_t^i)$, and observes the next state $\tilde s_{t}^i $. Then we can send the data $\{ (s_t^i, a_t^i, \tilde s_t ^i) , i\in [N]\}$ to any off-the-shelf policy evaluation solver to get the desired value function.
>
>
>
>
> Q2: Discussion of the trade-off between convergence rate and regularization error.
>
> A2: An excellent suggestion. We will add the following discussion: “Indeed,  it is worth emphasizing that the convergence rate to the regularized NE scales inverse proportionally with $\sqrt{\lambda}$, implying that convergence can be accelerated with a higher level of entropy regularization. On other hand, the approximation error of the regularized NE for the original unregularized NE scales proportionally with $\lambda$ (cf. the last equation on page 5). Therefore, it is desirable to choose the regularization parameter $\lambda$ that *balances the target accuracy level and convergence rate*.”
>
>
>
> Q3: Explain why introducing the mean embedding is necessary and helpful.
>
> A3: The mean-field state lies in the space of the distribution over $\mathcal{S}$, denoted by $\Delta(\mathcal{S})$, which is a continuous space. This makes it challenging to update the mean-field states. As shown in the paper “Learning Mean-Field Games”, to tackle such a challenge, a discretization method is proposed, where one maintains a $\epsilon$-net in the space of $\Delta(\mathcal{S})$ and project the updates to the $\epsilon$-net in each iteration. Such an updating step is both artificial and computationally expensive, and is only limited to the tabular case.
>
> When $\mathcal{S}$ itself is a continuous or infinite space, updating the mean-field states can be much for challenging. By adopting mean-embedding of the distribution, we can lift from the distribution to the RKHS, which can be viewed as an **infinite-dimensional feature representation** of the mean-field state. The key advantage is that now we can utilize via sample-based kernel regression to realize the update of mean-field states, which can be *efficiently computed*.  Utilizing mean-embedding, we can avoid realizing the updates of mean-field states via discretization and projection to $\epsilon$-net, which seems computationally intractable, and directly estimating the density function, which might require extra regularity conditions on the density associated with the mean-field state and the underlying state space $\mathcal{S}$.

---

> > ### Author Response · Authors · 2020-11-25
> > **Response to Reviewer 1 (Continued)**
> >
> > Q4: Why $\mathcal{M}$ is convex or why this is not an issue?
> >
> >
> > A4: $\mathcal{M}$ is convex, as it is the image of a linear mapping of the convex set $\Delta(\mathcal{S})$ (the probability simplex on $\mathcal{S}$).
> >
> > Q5: When the state space is infinite, either function approximation or non-parametric approach should be used for Q-function.
> >
> >
> > A5: It is a very good point. We agree that either function approximation or non-parametric method such as k-nearest neighbor should be used here. We will add additional discussion. As pointed out in and comment on Q1, line 3 in Algorithm 1 can be achieved by any data-driven policy evaluation algorithm as long as it outputs a statistically accurate value function. Besides KNN, other powerful function approximators such as RKHS and neural networks can also be utilized. See, e.g., [1] and [2].
> >
> > Thank you for other detailed comments. We will address these comments accordingly in the revision.
> >
> > [1] Regularized Policy Iteration with Nonparametric Function Spaces
> >
> > [2] Neural Temporal-Difference and Q-Learning Provably Converge to Global Optima

---

### Decision · Program_Chairs · 2021-01-07
**Final Decision**

**Decision:**

Reject

**Comment:**

This is interesting work, but not yet sufficiently mature for publication.  Although the authors propose an novel algorithm and provide an analysis, the reviewers raised several criticisms about the comparison to previous work, the lack of any empirical evaluation, the strength and unnaturalness of the assumptions used to establish convergence.  After discussion, the reviewers remained largely unsatisfied with the author responses to these questions, and none recommended accepting this paper.